# Accumulation by avalanches as significant contributor to the mass balance of a peripheral glacier of Greenland

Bernhard Hynek[1,2,3], Daniel Binder[4,3], Michele Citterio[5], Signe Hillerup Larsen[5], Jakob Abermann[2,3], Geert Verhoeven[6], Elke Ludewig[7], Wolfgang Schöner[2,3]

[1] Geosphere Austria, Department Climate Impact Research, Vienna, Austria

[2] Institut für Geographie und Raumforschung, Universität Graz, Austria

[3] Austrian Polar Research Institute, Vienna, Austria

[4] Institute for Geosciences University of Potsdam, Germany

[5] Geological Survey of Denmark and Greenland, Copenhagen, Denmark

[6] Department of Prehistoric and Historical Archaeology, Universität Wien, Austria

[7] Geosphere Austria, Sonnblick Observatory, Rauris, Austria

*Correspondence to*: B. Hynek (bernhard.hynek@geosphere.at)

*Keywords: UAV; structure from motion photogrammetry; Greenland; glacier mass balance; snow avalanches*

**Abstract.**

Greenland´s peripheral glaciers are losing mass at an accelerated rate and are contributing significantly to sea-level rise, but only a few direct observations are available. In this study we use the unique combination of high-resolution remote sensing data and direct mass balance observations to quantify the contribution of a singular avalanche event to the mass balance of Freya Glacier (74.38° N, 20.82° W), a small (5.5 km², 2021) mountain glacier in Northeast Greenland. Elevation changes calculated from repeated photogrammetric surveys in August 2013 and July 2021 show a high spatial variability, ranging from -11 m to 18 m, with a glacier-wide mean of $1.56 \pm 0.10$ m $(1.33 \pm 0.21$ m w.e.). After applying a seasonal correction of $-0.6 \pm 0.05$ m w.e. the geodetic mass balance over the entire eight-years period (2013/14 - 2020/21) is found to be $0.73 \pm 0.22$ m w. e. A significant influence on the near decadal mass balance stems from the exceptional winter mass balance of 2017/18, which was 2.5 standard deviations above average $(1.89 \pm 0.05$ m w.e.). After heavy snowfall in mid-February 2018, snow avalanches from the surrounding slopes affected more than one third of the glacier surface and contributed $0.35 \pm 0.04$ m w.e., which is close to 20% to the total winter mass balance of 2017/18. Remote sensing data show, that Freya Glacier is prone to avalanches also in other years, but to a lesser spatial extent. Due to a gap in mass balance point observations caused by high accumulation rates (buried stakes) and the COVID-19 pandemic the recently reported glacier-wide annual mass balance are rather crude estimates and show a negative bias of $-0.22$ m w.e. a$^{-1}$ compared to the geodetic mass balance. Finally, we speculate that the projected future warming may increase the likelihood of extreme snowfall, thus potentially increasing the contribution of snow avalanches to the mass balance of mountain glaciers in NE Greenland.

 **Graphical Abstract.**

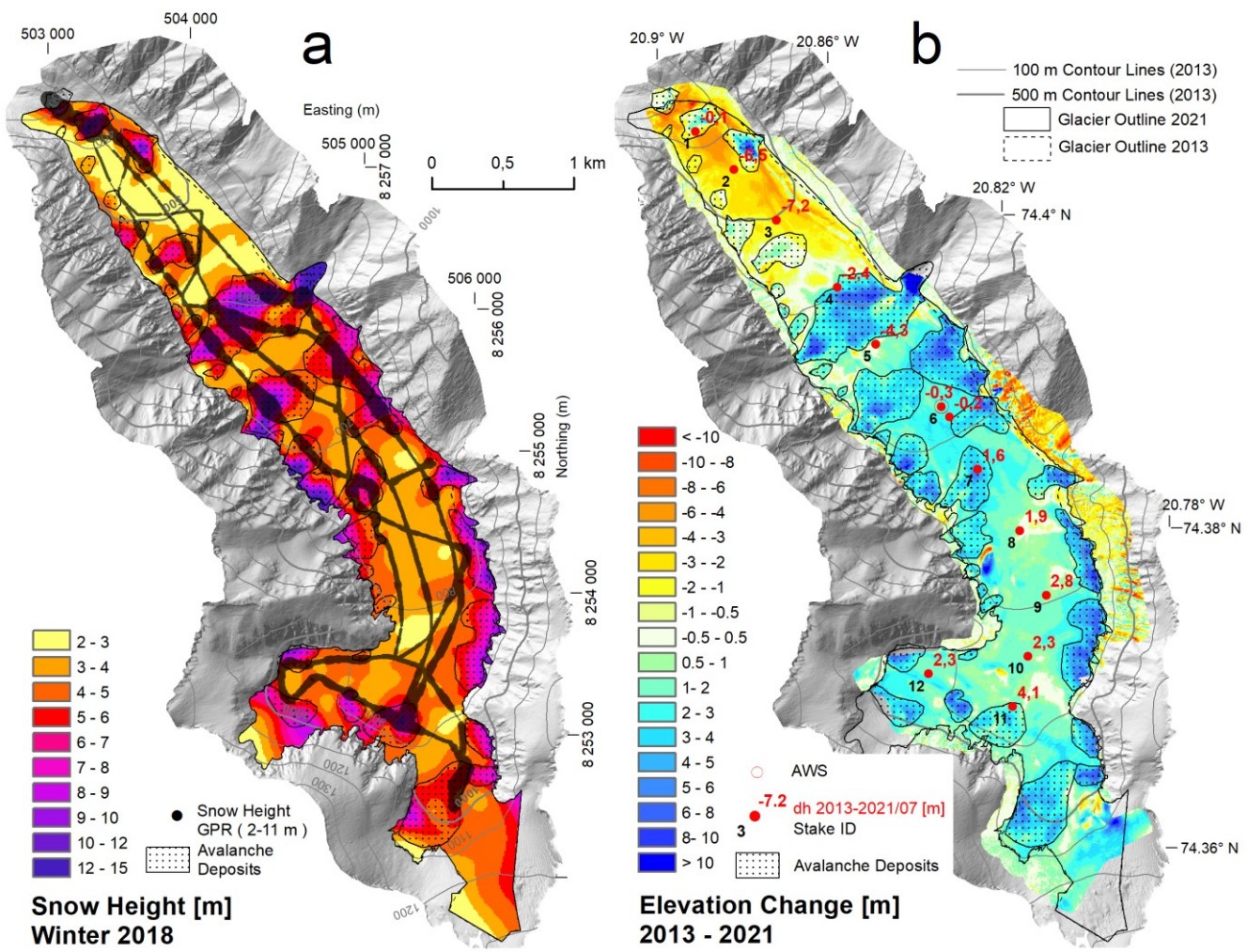

a) Measured (GPR) and extrapolated snow height in winter 2018 and delineation of avalanche affected areas. b) Elevation Change between 18.8.2013 and 27.7.2021 and measured ablation at the stake locations.

## 1 Introduction

The ice cover of Greenland consists of the Greenland Ice Sheet and approximately 20 300 peripheral glaciers (Abermann et al., 2019b; Rastner et al., 2012). Although Greenland´s peripheral glaciers comprise only 4% of the total ice cover of Greenland, their recent contribution to mass loss from Greenland (11%) and global sea-level rise is disproportionately high compared to that of the ice sheet (Khan et al., 2022). This confirms their higher sensitivity to current climate change. During the last 60 years mass loss from Greenland's peripheral glaciers comprise ~ 8% of the world's land ice contribution to sea-level rise (Frederikse et al., 2020; Zemp et al., 2019).

While the overall mass loss from Greenland's peripheral glaciers has accelerated during the last two decades, the pattern is heterogeneous on a regional scale (Hugonnet et al., 2021). In Northeast Greenland, specifically, the mass loss has decelerated, with continued thinning at lower elevations and thickening at higher elevations (Khan et al., 2022). The decelerated mass loss in Northeast Greenland has been associated with an increase in precipitation (Hugonnet et al., 2021), whereas the reduced mass loss of Icelandic and Scandinavian glaciers, for example, has been associated with North Atlantic cooling (Noël et al., 2022).

However, our knowledge of the individual drivers of mass changes of Greenland's peripheral glaciers is limited as direct observations and process studies are scarce. Machguth et al. (2016b) compiled all reported mass balance observations in Greenland and showed that while mass balance observations on the ice sheet have increased tenfold, the peripheral glaciers are still heavily undersampled despite their topographical and climatological complexity. To our knowledge, currently only 6 out of 20 300 glaciers and icecaps in Greenland are monitored (Abermann et al., 2019b). Three of these are located on the 2600 km long east coast: Mittivakkat Glacier on Ammassalik Island (65° N) (Mernild et al., 2013; Yde et al., 2014), A. P. Olsen Ice Cap (Citterio and Ahlstrøm, (2010); Larsen et al., (2023) and Freya Glacier (both at 74° N near Zackenberg Research Station).

The mass balance monitoring at Freya glacier has been carried out using the direct or glaciological method (Kaser et al., 2003; Østrem and Brugmann, 1991) which is based on various point observations of ablation and accumulation distributed over different elevations on the glacier. These point observations of mass change are then extrapolated to estimate the annual mass balance of the entire glacier, often incorporating additional information such as the position of the snowline. However, the specific implementation of this step may vary among glaciers and observers (Zemp et al., 2013) and also depends on the number and distribution of available point measurements. Annual mass balance measurements are likely to accumulate systematic errors over the years (e.g. Huss et al., 2009), therefore it is recommended to compare and, if necessary, homogenise the annual mass balance time series using decadal volume changes based on geodetic surveys of the glacier surface (Huss et al., 2009; Klug et al., 2018; Zemp et al., 2013). On Freya Glacier these geodetic surveys were carried out in 2013 and 2021 using an Image-Based 3D surface Modelling (IBM) approach.

In the last decade, hybrid photogrammetric computer vision-based approaches have become commonplace in many academic fields. With photogrammetric methods at their core, these hybrid approaches mainly rely on the computer vision algorithms Structure from Motion (SfM) and Multi-View Stereo (MVS) to digitally extract three-dimensional (3D) surfaces from overlapping images. These 3D surfaces can then be used to produce accurate orthophotos. Often, such SfM-MVS approaches utilize terrestrial photographs acquired with consumer-grade cameras (Piermattei et al., 2015; Marcer et al., 2017) or images obtained via cameras mounted on uncrewed aerial vehicles (UAVs) (e. g. Gindraux et al., 2017; Rossini et al., 2018; Geissler et al., 2021).

Interestingly, there are only a few studies on the contribution of snow avalanches to the mass balance of glaciers despite the apparent importance of this accumulation process. Glaciers with considerable accumulation from avalanches have been associated with high and steep headwalls typical for High Mountain Asia (Laha et al., 2017). (Kneib et al., 2024b) showed, that a lot of glaciers in the European Alps are also avalanche fed. In the Arctic, rising temperatures may increase the number and intensity of snowfall events as observed over NE Greenland in 2018 (e.g. Schmidt et al., 2019) which will in turn enhance avalanche activity (Abermann et al., 2019a). However, the contribution of avalanches to the mass balance of individual glaciers is difficult to measure, therefore it has been quantified by applying precipitation factors locally at the base of headwalls to fit the observed ice flux (Laha et al., 2017; (Kneib et al., 2024a; Laha et al., 2017).

This study examines the effects of an extraordinary winter accumulation combined with widespread avalanche activity on the mass balance of an High Arctic mountain glacier. In particular, we quantify the contribution of avalanches to the winter mass balance 2017/18 of Freya Glacier by taking advantage of a detailed ground penetration radar survey of snow depth conducted in April 2018. Furthermore we demonstrate the imprint of avalanches in high-resolution glacier elevation changes 2013 - 2021.

We calculate IBM-derived elevation changes and deduce the geodetic mass balance of Freya Glacier between 2013/14 and 2020/21. We delineate snow avalanche deposits from February 2018 on the glacier area, quantify their mass contribution to the winter mass balance 2017/18 and show their imprint on the multi-year geodetic mass balance. Finally, we compare the geodetic mass balance to the cumulative glaciological mass balance, discuss likely error sources for the discrepancy and emphasise the need for a reanalysis of the glaciological record. This need arises due to the observational gaps caused by travel restrictions during the COVID-19 pandemic and a limited observational network that proved insufficient to account for the recent spatial variability of surface mass balance on the glacier.

## 2 Freya Glacier

Freya (Freja, Fröya[i]) Glacier (74,38° N, 20.82° W) is a polythermal mountain glacier (Binder et al., 2009) located on Clavering Island in Northeast Greenland, 10 km southeast of Zackenberg Research Station (Fig. 1). The coastal glacier is oriented towards the Northwest, surrounded by steep ridges on both sides, spans an elevation of 1300 m to 280 m a.s.l. and covers a surface area of 5.5 km² (2021). The glacier was subject to glaciological investigations already in the late 1930s (Ahlmann, 1942, 1946) likely due to its relatively good accessibility. During the International Polar Year 2007/2008 a mass balance monitoring programme was initiated (Schöner et al., 2009) and has been ongoing since (Hynek et al., 2014; World Glacier Monitoring Service (WGMS), 2023). The current monitoring consists of a stake network, an automatic weather station (AWS) of the PROMICE setup (Fausto et al., 2021) and two high-quality webcams (Hynek et al., 2018). Daily images from the two webcams are publicly available via the websites foto-webcam.eu (Freya Glacier Webcam 1: https://www.foto-webcam.eu/webcam/freya1/ and Freya Glacier Webcam 2: https://www.foto-webcam.eu/webcam/freya2/[1]).

---

[1] Due to technical problems the webcams are offline since April 2024, but older camera images can still be found here.

## 3 Data and Methods

### 3.1 Geodetic Survey 2013

Due to the ease of the process and the suitable topography, SfM-MVS-based image-based 3D surface modelling was the optimal choice for generating a DEM of Freya Glacier during the 2013 field campaign. Although no UAV was available, the ridges around the glacier provided useful natural viewpoints for a ground-based survey. Between 11[th] and 18[th] August 2013, we took oblique overlapping photographs of the glacier surface from about 450 locations on the slopes on both sides of the glacier using a Nikon D7100 digital single lens reflex camera with a 20 mm fixed lens. Simultaneously with the image acquisition, we surveyed approximately 100 natural Ground Control Points (GCPs) using a differential GNSS (Global Navigation Satellite System) receiver (Fig. 2 a-c). For post-processing of the survey, a temporary GNSS reference station was established on stable rock next to the glacier. We surveyed the upper part of the glacier on the 11[th] and 12[th] of August 2013, when the glacier surface was almost snow-free. Snowfall on 14[th] August followed by a period of low visibility marked the end of the melt season. On 18[th] August 2013, we surveyed the lower part of the glacier. Surface ablation between the survey dates was below 0.15 m and was partly compensated by an average fresh snow height of 0.10 m.

### 3.2 Geodetic Survey 2021

The second high-resolution DEM used in this study stems from 2021. On 29[th] and 31[st] July 2021, we used a UAV (DJI Phantom 4 RTK) to obtain an overlapping image series of the glacier surface. On 29[th] July, we photographed 80% of the glacier surface (lower part) and finished the drone flights on 31[st] of July. On 28[th] and 29[th] of July 2021, we surveyed approximately 100 mainly artificial GCPs on the glacier surface using a differential GNSS receiver and a base station that was put up at the same location as in 2013 (Fig. 2 d-f). During the survey, surface ablation between 28[th] and 31[st] July was less than 0.2 m. Table 1 lists the main characteristics of both photogrammetric surveys.

### 3.3 GNSS and IBM workflow

GNSS raw logs containing the GCPs and the UAV trajectory were post-processed using the reference station next to the glacier. Coordinates were transformed into UTM coordinate reference system (zone 27N, epsg:32627) and to orthometric heights (egm96). For the accuracy assessment of the surface reconstruction, one subset of the GCPs was used to reference the generated 3D model (control points), and another subset was used to validate the 3D model (independent check points). All GCPs were used to reference the final DEM. GCPs that were not clearly visible in the imagery were used for elevation validation of the final DEM output. The workflow of the DEM and orthophoto generation followed the classical SfM process (e.g. Rossini et al., 2018) using Agisoft Metashape (AgiSoft LLC, 2023). Due to the different surface texture (snow covered vs snow free) of the lower and upper 2013 imagery, these parts of the glacier were processed independently and combined to one final DEM afterword (see supplement).

### 3.4 Elevation Changes

Elevation changes between 2013 and 2021 were calculated by DEM differencing in 1 m planar resolution. As the georeferencing of the two final DEMs is based on a large number of GCPs, a co-registration of the DEMs (Nuth and Kääb, 2011) was not necessary. Elevation differences in overlapping ice-free terrain had a mean bias of 0.1 m and a standard error of 0.45 m (see supplement). Most of the likely stable terrain is rather steep, and in some areas the DEM 2013 might have larger errors than everywhere else, so we did not correct for this bias.

### 3.5 Density Assumption and Geodetic Mass Balance

To convert the observed volume change into a mass change we use the conversion factor of $850 \pm 60$ kg/m³ recommended by Huss (2013) for periods longer than 5 years, with stable mass balance gradients, the presence of a firn area and volume changes significantly different from zero. No firn density measurements have been carried out on the glacier so far, neither in the accumulation zone nor in one of the avalanche deposits. The main part of the accumulation that led to the observed positive elevation changes occurred in 2018 and has undergone densification over four melt seasons by the time of the second survey. However, percolation and the possible formation of ice lenses might create high variability in firn density (Vandecrux et al., 2018, Machguth et al., 2016a). Therefore we decided to follow the recommendation of Huss (2013). In 2013 the survey was very close to the end of the ablation season. In 2021 an adjustment of $-0.6 \pm 0.05$ m w.e. was calculated between the survey on 29[th] July and the end of the ablation season on 5[th] September based on 10 ablation stake readings.

### 3.6 Glaciological mass balance

#### 3.6.1 Winter mass balance

Due to logistical challenges in accessing the glacier with a snow mobile, the number of snow height observations varies considerably from year to year. Distributed winter snow height is measured either by 40 - 150 manual snow depth probings, or by a 800 MHz GPR snow survey of several km in length. In April 2018, an extended GPR snow survey with a total length of 27 km was carried out to capture the spatial distribution of snow depth including the still visible avalanche deposits. To get a regular grid of snow height, a spline function was fitted to the data. In contrast, snow density was measured at only one location, which was not influenced by avalanches: in a snow pit next to the AWS at an elevation of 680 m (Fig. S4) . Winter mass balance was calculated as a spatial average of snow depth over the whole glacier area multiplied with an extrapolated snow density based on the available measurement next to the AWS. Similar GPR snow surveys with comparable spatial coverages have been carried out in spring 2008 and 2017, while in other years a reduced sample network was used.

#### 3.6.2 Annual mass balance

Until 2015 annual mass balance measurements were usually carried out in August, and seasonal mass balance was measured at several points distributed across the glacier. Annual glacierwide mass balance was then determined by extrapolating the point values onto the whole glacier area. Depending on the number of point observations the mean standard error is estimated as 0.05 m w.e. a$^{-1}$ (e.g. Pulwicki et al., 2018). Mainly due to high travel costs, but also in accordance with the mass balance monitoring at A. P. Olsen Ice Cap the monitoring strategy was changed in 2016 to only one visit per year in spring. At the same time an automatic monitoring system was installed, namely an automatic weather and mass balance station and an automatic camera to track the retreat of the snow line during summer. Since then annual mass balance has still been measured at eleven ablation stakes, which usually protrude from the winter snow. At each stake the mass balance of the previous year is determined by measuring the current snow depth and the height change of the stake. However, because of above average snow heights in spring 2018 and 2019 only two stakes were found. In 2020 and 2021 spring measurements were not possible due to the travel restrictions caused by the COVID-19 pandemic. Therefore, the glacierwide mass balance from 2016/2017 to 2020/21 was reconstructed using a linear relationship (see supplement for details) between the mass balance at the AWS (index stake) and the glacierwide mass balance based on observations from 2008 to 2016, introducing an estimated uncertainty of 0.2 m w.e. a$^{-1}$. Most stakes were found again and could be measured in July 2021 and April 2022.

### 3.7 Quantifying the influence of avalanches on the winter mass balance of 2018

To delineate the avalanche deposits of 2018 we identified areas with a strong increase of snow heights along the GPR tracks. In areas without GPR tracks, we completed the delineation using a best estimate based on pictures of avalanche cracks, remnants of avalanches in the orthophoto of 2021, above average local elevation changes 2013-2021 and likely avalanche flow paths based on topography. The GPR snow depth data was sampled down to 10 m point distance and then interpolated using a spline function onto a grid of glacier-wide snow heights. To estimate the contribution by avalanches to the winter mass balance of 2018, we calculated spatial averages of the snow height grid in avalanche affected areas and in avalanche free areas. To convert snow heights into snow water equivalent, we used the mean snow density of 385 kg m$^{-3}$ (measured in the snow pit next to the automatic weather station) for areas that are not influenced by avalanches. As snow density typically increases with snow depth and avalanche deposits have higher snow densities than the undisturbed snow pack we used higher snow densities for the avalanche deposits: a 5% increased snow density (404.25 kg m$^{-3}$) as a best guess and 10% increase (423.5 kg m$^{-3}$) which we interpret as an upper boundary.

### 3.8 Climate data

Snow height at the AWS on Freya Glacier is measured by two Campbell SR 50 ultrasonic devices, one fixed to the mast of the weather station 3.4 m above the ground and one fixed to an ablation stake. Both sensors were buried in snow by mid-February 2018. On 28$^{th}$ April, the weather station was reestablished on the surface (Fig. 3). The data gap of 2.5 months was reconstructed using snow height data from the main weather station at A. P. Olsen Ice Cap (Larsen et al., 2023; Greenland Ecosystem Monitoring, 2020a), which has a continuous record in 2018 (see supplement, Fig. S6). Additionally, we used temperature data from the climate station Zackenberg (Greenland Ecosystem Monitoring, 2020b) and precipitation data from the ERA5 global reanalysis (Hersbach et al., 2020).

### 4. Results

### 4.1 DEM and orthophoto 2013

The shaded relief of the 2013 DEM (Fig. 4a) shows a high level of detail, with only a few artefacts visible in the middle and uppermost part of the glacier. These artefacts occur, where the distance between the photo points and the glacier surface is high and the angle towards the glacier surface is acute. The middle part of the glacier is poorly covered, the GCPs there (Fig 4a, set 3) could not be identified in the images and were used to check only the vertical accuracy of the DEM in that area (Table 2). The orthophoto shows almost snow-free conditions in the upper part of the glacier and the new snow on the lower part (Fig. 4b). The surface reconstruction covers the entire glacier area and the adjacent ridges. Since all GCPs are on the glacier surface, the accuracy of the surface reconstruction is expected to drop significantly in the adjacent ridges. The accuracy of the surface reconstruction expressed as RMSE at the check points is significantly worse than the RMSE at the control points, with lateral accuracy being particularly poorer than the vertical accuracy (Table 2).

## 4.2 DEM and orthophoto 2021

The shaded relief of the 2021 DEM (Fig. 5a) shows a much higher level of detail due to better measurement geometry and resolution. The ground sampling density (Table 1) and the accuracy of the surface reconstruction (Table 2) of the 2021 survey are both higher than those for the 2013 survey. However, only 95% of the glacier surface is reconstructed, and the DEM does not extend much to the adjacent ridges due to limited UAV battery supply during fieldwork. Avalanche affected areas are visible in the orthophoto (Fig. 5b) on the lower and middle part of the glacier, while the upper part was still covered by slush and winter snow.

## 4.3 Elevation Changes and Geodetic Mass Balance

Elevation changes in 1 m resolution (Fig. 6) were calculated for 95% of the glacier area, missing only some parts in the upper accumulation zones and a small debris covered area next to the glacier snout. Elevation changes for these areas were calculated by fitting a spline function to the elevation changes in the surrounding areas, to avoid a bias in the geodetic mass balance. Elevation changes show a high spatial variability. Surface lowering is observed on 20% of the glacier surface, mainly at elevations below 600 m a.s.l., and reaching a minimum of -11 m in the lowest part of the glacier. Above 600 m a.s.l. elevation changes are mainly positive. At the centerline of the glacier, elevation gains are mainly smaller than 2 m. In several distinct areas predominantly along both sides of the glacier, elevation gains are up to several meters with a maximum of 17 m. These areas coincide with potential avalanche depositions from large side valleys (Fig. 7). The mean elevation change from August 2013 to July 2021 for the entire glacier is $1.56 \pm 0.10$ m. The main sources of uncertainty include ablation during the survey, unmeasured areas, and the uncertainty in the delineation of the glacier surface area. Converting this volume change into a mass change – and hereby introducing another uncertainty using a density assumption of $850 \pm 60$ kg/m³ – we obtain the specific geodetic mass balance from August 2013 to July 2021 as $b_{geod} = 1.33 \pm 0.21$ m w.e. After accounting for the mass losses during August 2021, the total 8-year geodetic mass balance 2013/14 - 2020/21 adds up to $b_{geod.8y} = 0.73 \pm 0.22$ m w.e. (0.09 m w.e. a$^-$$^1$).

## 4.4 Winter 2018 and avalanches

In the winter of 2017/2018, a series of low-pressure systems between the southern tip of Greenland and Iceland transported warm and moist air masses to the East Coast of Greenland with frequent snowfall leading to above average winter precipitation sums along large parts of the East Coast (Fig. 8) and also on Freya Glacier (Fig. 9). Between 12[th] and 18[th] February 2018 approximately 1.5 m of snow accumulated within five days on Freya Glacier. This led to widespread avalanche activity, and during fieldwork in April 2018, signs of large avalanche deposits were visible across the entire glacier. Particularly in the middle part of the glacier, several large avalanches originating from the tributary valleys on both sides covered large parts of the glacier. In April 2018, avalanche deposits were found on 36% of the glacier area. Individual GPR-derived snow heights ranged from 2.2 m up to 12.1 m, with a mean snow height of 4.4 m. The pattern of snow height distribution in winter 2018 and the elevation changes show a high similarity (Fig. 6). The area-averaged snow height on the entire glacier is 4.8 m, with 6.2 m on avalanche deposits, and 4.0 m in areas without avalanches. The snow height contribution from avalanches averaged over the whole glacier is 0.8 m. Mean snow density at the snow pit next to the AWS at stake 6 was 385 kg/m³. We consider this as a lower limit for the glacier-wide spatial mean snow density as avalanche snow likely has a higher density than the undisturbed snow cover in the middle of the glacier (Sovilla et al., 2001), where the snow density measurement was carried out. Assuming a 5% higher snow density on the glacier due to compaction and overburden pressure within the avalanche deposits, the specific mass balance

contribution of avalanches is 0.35 $\pm$ 0.04 m w.e., which accounts for 19% of the total winter mass balance of 1.89 $\pm$ 0.05 m w.e. (Table 3).

265

## 4.5 Visibility of avalanches on the glacier surface

While remnants of snow avalanches have been visible on the glacier surface over several years, particularly between 2012 and 2016 (Fig. 10, Fig.11), their surface extent was smaller compared to 2018. In 2016 at least three avalanche deposits are visible on the orographic right side of the glacier in orthophotos taken in July 2016, and some more can be identified in August 2016 270 (Fig. 10).

## 4.6 Glaciological mass balance

The time series of winter and annual mass balances (World Glacier Monitoring Service, 2022) of Freya Glacier are shown in Fig. 12. Prior to 2013, all annual mass balances were negative, with 2013 having the most negative mass balance on record so 275 far. Higher winter mass balances between 2014 and 2018 can be associated with some positive annual mass balances in that period, while after 2019 drier winters facilitated again negative annual mass balances. Especially stake 1 and stake 4 are influenced by avalanches and show reduced ablation rates (see stake readings in Fig. 6b and supplement, Table S1). The cumulative glaciological mass balance 2013/14 - 2020/21 is -1.0 $\pm$ 0.4 m w.e. The bias with respect to the geodetic mass balance is -1.73 m w.e. or -0.22 m w.e. a$^{-1}$.

**5 Discussion**

The avalanche cycle of 2018 was outstanding in regard to the mass input and the glacier area affected. However, avalanches seem to be a persistent feature on Freya Glacier, as their deposits are visible almost every year. It is difficult to date these avalanches and estimate their frequency, as older avalanche deposits might get covered by new ones. In case of the two big 285 avalanches in the middle of the glacier which originated from opposite sides and travelled almost all the way through the other side of the glacier in 2018 (green circle in Fig. 11), we have strong evidence, that their remnants are still visible in the orthophoto of 2021, more than three years after the incident. On the one hand it takes a few ablation seasons to melt avalanche snow up to 8 meters thick, particularly, if that snowpack is located in a rather flat area on the glacier, where it is more likely to get densified by retention of meltwater. On the other hand, winter mass balances in the following years were below average (Fig. 12) and 290 therefore unlikely to produce avalanches of this size. While avalanche deposits are easy to identify in the ablation area or in rather negative mass balance years, their presence and extent remain equivocal in the upper firn area or less negative years.

A major uncertainty in the geodetic mass balance is introduced by the density assumption. On Freya Glacier, high accumulation rates by avalanches have generated thick and possibly dense firn layers with high potential of meltwater retention and refreezing. 295 First, it is difficult to constrain the initial snow density of snow deposited by avalanches without a direct measurement within these deposits. Li et al. ( 2021) and Sovilla et al. (2001) observed that the snow density of avalanche deposits might be two to three times higher than the undisturbed snowpack at the time of the avalanche release date. As a best guess, we used 5 to 10% higher snow density within the avalanche deposits compared to the undisturbed snow cover to calculate the winter mass balance. Second, there is a large uncertainty, how the density of these avalanche deposits evolves during the following four years, as this 300 is mainly influenced by the possible formation of impermeable ice layers by percolating meltwater, as shown by Braithwaite et

al. (1994) and Vandecrux et al. (2018). Also Machguth et al. (2016a) showed that firn loses a part of its capacity to store water after forming near-surface ice layers during strong melt events. While we observed strong melt events on Freya Glacier during the following sommers, refreezing of meltwater has already been suspected to a play an important role in the mass balance of Freya Glacier (Ahlmann, 1946) and was observed qualitatively during fieldwork in 2021. The bright glacier surfaces, that are

305 the supposed remnants of avalanches, looked like snow, but proved to be as hard as ice. Given all these uncertainties and the lack of firn density measurements at Freya Glacier, we chose to stick to the recommendation of Huss (2013), who showed in a model experiment that a conversion factor between elevation change and mass change of $850 \pm 60$ kg m$^{-3}$ is appropriate for a wide range of conditions over longer time periods.

The cumulative glaciological mass balance for the period 2013/14 – 2020/21 was estimated in a rather crude way and carries uncertainties for several reasons. The accumulation within the avalanche deposits visible in the satellite images from 2014 to 2016 might have been underestimated. From 2017 to 2021, only one or two point observations were available, so the glacierwide mass balance was reconstructed using a linear relationship based on the mass balance at the AWS (stake 6). Another likely reason for the bias between the glaciological and geodetic mass balance is the already mentioned unknown magnitude of

meltwater retention by refreezing within deeper firn layers. This process is difficult to measure; in our case it was not feasible to measure firn density due to logistical reasons. A thorough reanalysis of the annual mass balance series using all available data and following a methodology based on Zemp et al. (2013) is necessary, but beyond the scope of this paper.

Regardless of the recent uncertainty in the glaciological mass balance time series of Freya Glacier there is a shift from rather

negative to less negative mass balances starting in 2013/2014, which we attribute to higher winter accumulation between 2014 and 2018. This shift to less negative mass balances – caused by an increase in precipitation over NE Greenland in recent years – has been shown to be a regional effect by Hugonnet et al. (2021) and Khan et al. (2022).

## 6 Conclusions

Our study shows that the 8-year geodetic mass balance of Freya Glacier from 2013/14 to 2020/21 has been positive (0.73 + 0.22 m w.e.). A significant positive contribution to the mass balance stems from avalanches originating from the surrounding slopes. While avalanche deposits are visible on the glacier surface to a limited extent almost every year, the winter 2018 was clearly outstanding. After a heavy precipitation event in mid-February 2018, which caused a snow height increase of 1.5 m within 5 days, widespread avalanche activity affected more than one third of the glacier area. Based on a detailed GPR

survey conducted in April 2018, we estimated the contribution of avalanches to the winter mass balance of 2018 to be 0.35 + 0.04 m w.e., which is close to 20% of the winter mass balance of $1.89 \pm 0.05$ m w.e. We showed that some avalanche deposits are visible on the glacier surface almost every year, leaving a strong imprint on the elevation changes. A main uncertainty in this assessment arises from a lack of snow and firn density measurements, particularly within the avalanche deposits, but also in the upper firn areas. The cumulative glaciological mass balance for 2013/14 to 2020/21 is negative (-1.0 + 0.4 m w.e.),

suffering from data gaps and only a few point observations in recent years. The magnitude of the bias between geodetic and glaciological mass balance (-0.22 m w.e. a$^{-1}$) is similar to bias estimates reported by Andreassen et al., (2016) for ten glaciers in Norway and therefore as such not unexpected (see also Zemp et al., 2013). Likely reasons for this bias include the underestimation of the mass contribution by avalanches, the general lack of distributed accumulation measurements, and possibly the underestimation of refreezing meltwater leading to internal accumulation. Capturing these processes, as well as

firn density measurements, should receive more attention in future mass balance monitoring at Freya Glacier. Assuming a

higher likelihood of strong winter precipitation events in a warmer climate, we expect accumulation by avalanches to become more important on Arctic mountain glaciers that are situated in or surrounded by steep terrain.

**Data availability**

Mass balance data of Freya Glacier are available through the WGMS (wgms.ch) and pangaea.de. The DEMs and orthophotos of 2013 and 2021 have been submitted to pangaea.de. Data which are not yet available there, can be requested from the authors.

**Author Contributions**

BH designed the study, conducted the data analysis and wrote the manuscript. BH, DB carried out the geodetic surveys. BH, DB, MC, SHL, JA and WS carried out mass balance observations on Freya Glacier. DB analysed the GPR data. GV helped with planning and processing of the 2013 geodetic survey. WS and EL provided the funding. All authors provided insights regarding the interpretation of data and reviewed and edited the manuscript.

The authors declare that they have no conflict of interest.

**Acknowledgements**

The work is supported by the International School for Alpine Research at Sonnblick Observatory (ISAR-SBO), a research Grant from GeoSphere Austria. The survey in 2013 was supported by Österreichische Gesellschaft für Polarforschung. Data from the Greenland Ecosystem Monitoring Programme were provided by the Geological Survey of Denmark and Greenland (GEUS),
Denmark and Asiaq – Greenland Survey, Nuuk, Greenland. The authors are grateful to the logistics team at Zackenberg Research Station for the logistical support of the fieldwork. The authors gratefully acknowledge Geo Boffi for post-processing the GNSS data of 2013, Anders Anker Bjørk for taking overlapping imagery from an airplane in August 2016 and Anna Rohrböck for the plotting ERA5 data. Comments and suggestions of two anonymous referees and the scientific editor Etienne Berthier substantially improved the paper.

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

**Figures and Tables:**

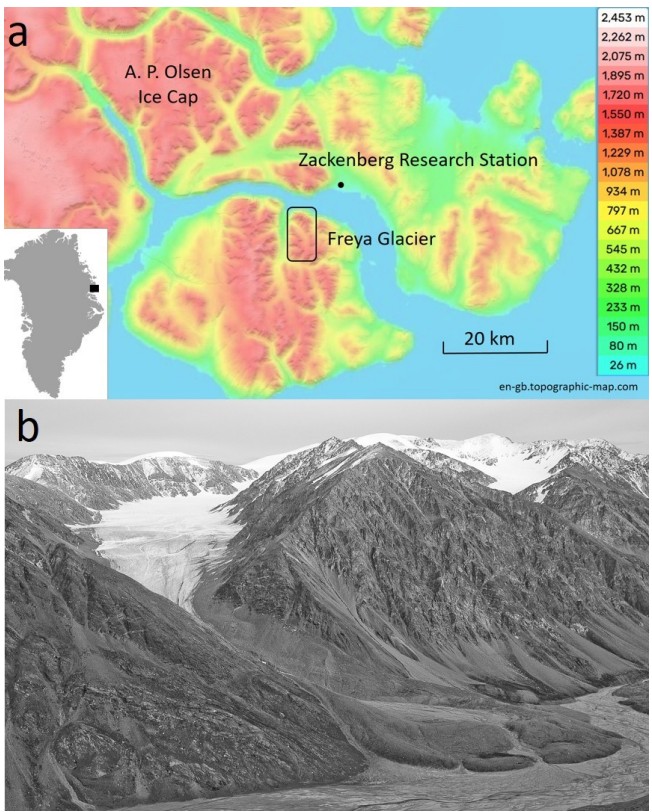

**Figure 1: a) Location of Freya Glacier (74.38°N, 20.82°E) on Clavering Island in Northeast Greenland, next to Zackenberg Research Station and A. P. Olsen Ice Cap. (Map from en-gb.topographic-map.com) b) Picture of Freya Glacier and its surrounding ridges in August 2008 (Photo: B. Hynek).**

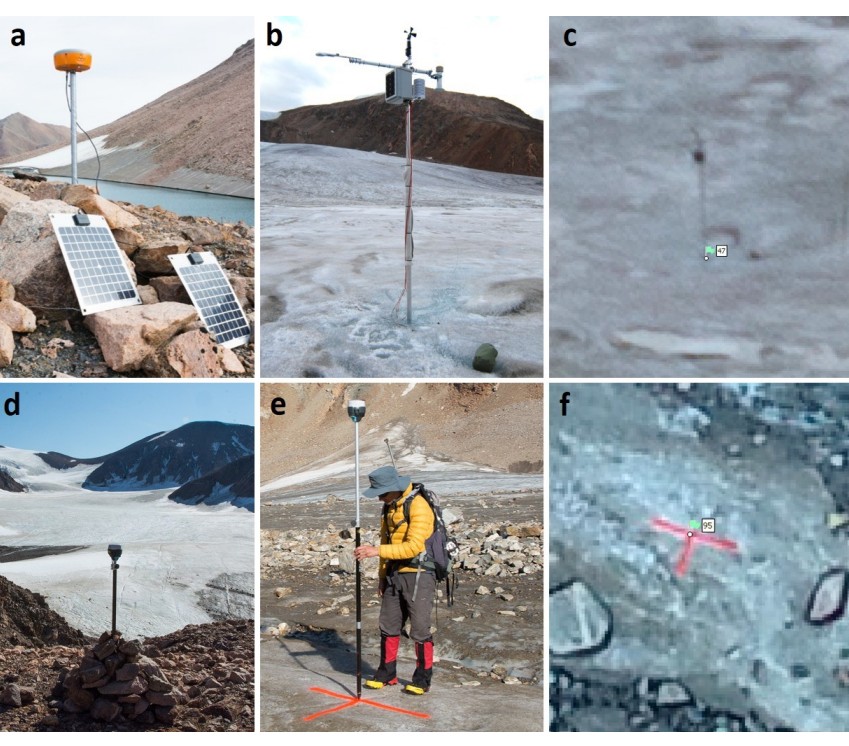

**Figure 2: Upper panel: GNSS Survey 2013. a) GNSS base station b) example of a natural GCP and c) its visibility in the imagery. Lower panel: GNSS Survey of 2021. d) GNSS Base Station e) Survey of an artificial GCP and f) the visibility of the GCP in the imagery.**

**Table 1:** Main characteristics of the two SfM-MVS surveys.

| | 2013 | 2021 |
|---|---|---|
| Survey dates | 11. - 18.8.2013 | 27.-31.7.2021 |
| Survey Geometry | Oblique (Terrestrial) | Nadir (UAV) |
| Camera/UAV | Nikon D7100 + 20mm | Phantom 4 RTK |
| Image Resolution | 24 Mpix | 20 Mpix |
| No of Images | 430 | 6250 |
| Height above glacier surface | 10 - 400 | 140 |
| Ground Sampling Distance | > 20 cm | 3.8 cm |
| No. of visible GCPs | 67 | 68 |
| Density of visible GCPs [ /km²] | 12.6 | 13.6 |
| Max. elevation change during survey [m] | < 0.15 | < 0.20 |
| Surface reconstruction [% of Glacier Area] | 100% | 94% |
| DEM spatial resolution [m] | 1 | 0.2 |
| Orthophoto spatial resolution [m] | 0.25 | 0.05 |

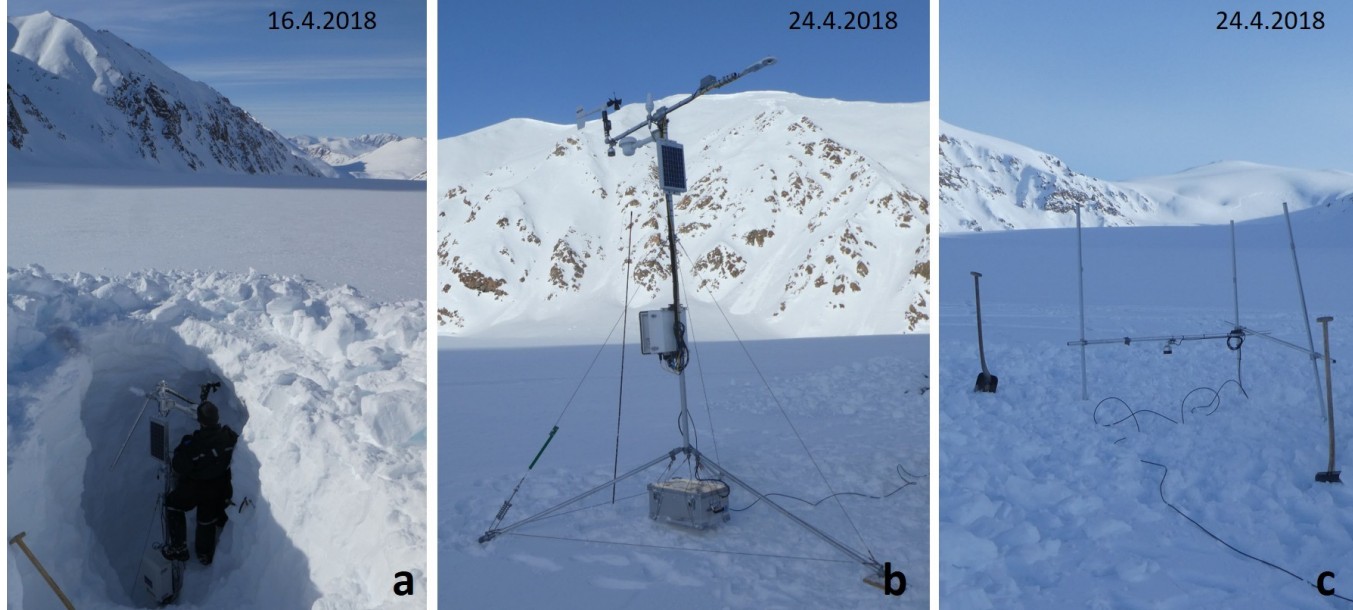

**Fig. 3: Maintenance of the AWS at Freya Glacier in April 2018: a) The station is 3.5 m tall and was completely covered in snow. b) The weather station and the c) stakes with the second ultrasonic device were re-established on the snow surface. (Photos: Daniel**
**Binder).**

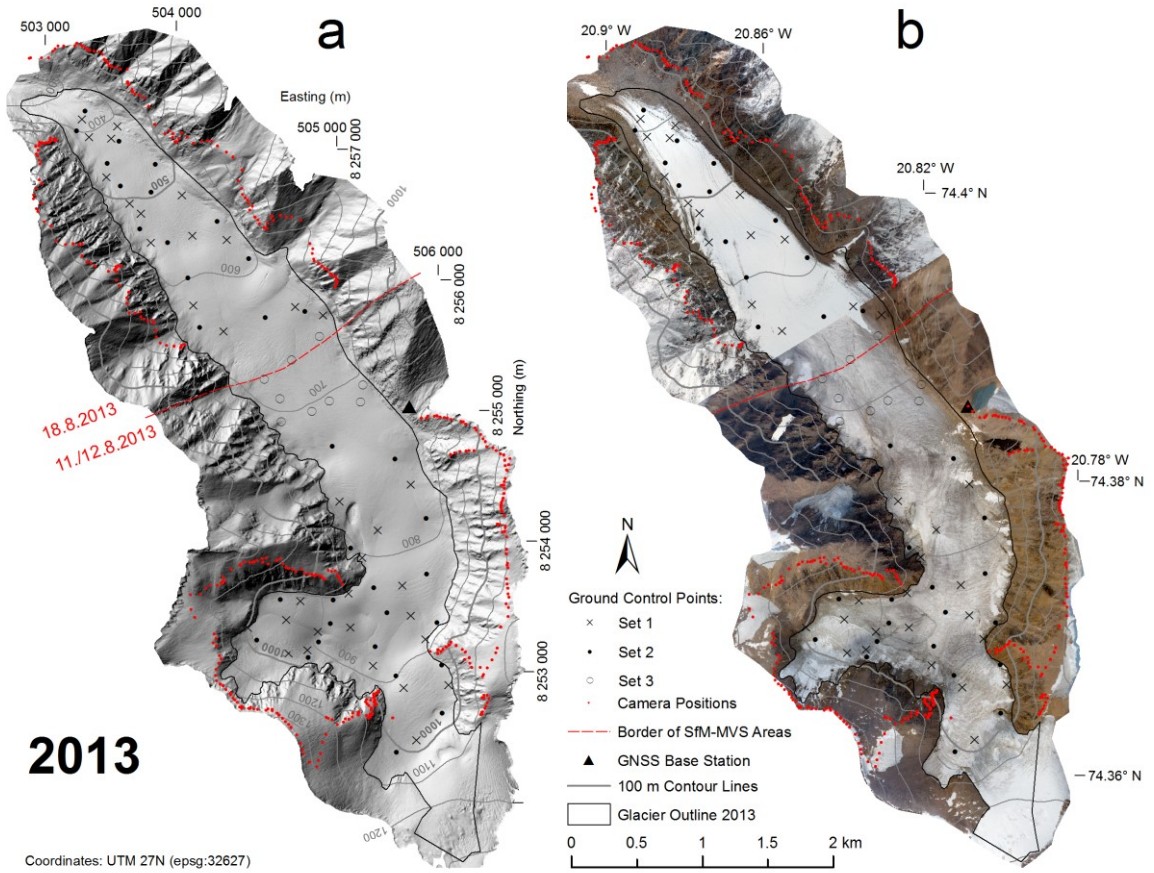

**Figure 4: a) Hillshade of the resulting DEM 2013 in 1 m resolution and b) Orthophoto of the survey in August 2013. On both maps the locations of the photo points, the ground control points (GCPs) and the GNNS Base Station are indicated. The upper part of the glacier was surveyed on 11.8. and 12.8. The lower part of the glacier was surveyed on 18.8., after snowfall marked the end of the ablation season 2013.**

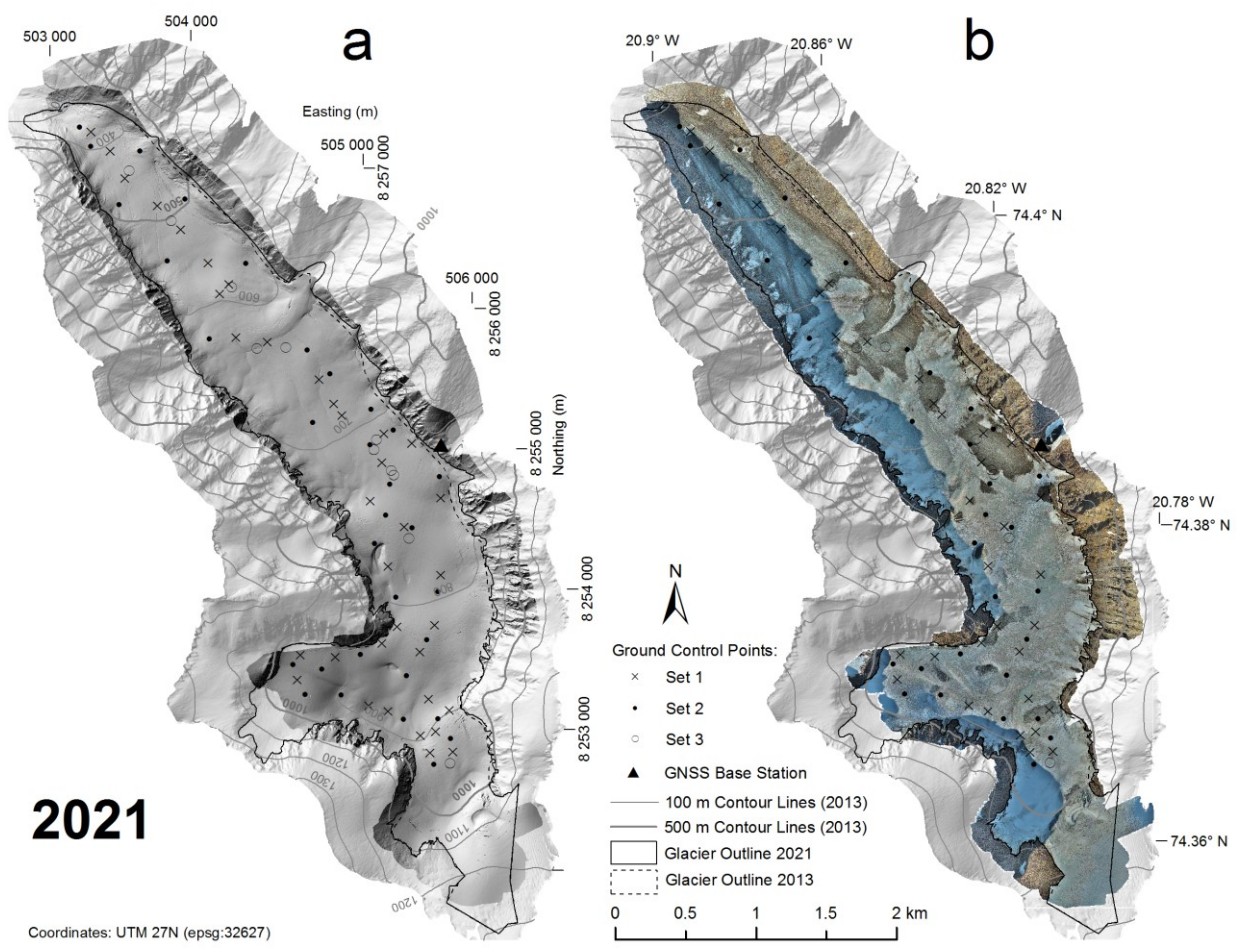

Figure 5: a) Hillshade of the 2021 DEM (dark grey) in 1 m resolution and b) Orthophoto of the survey in July 2021. On both maps the hillshade of 2013 is displayed in the background and the locations of the ground control points (GCPs) and the GNSS base station are indicated. The lower part of the glacier was photographed on 27.7.2021 and the upper part on 31.7 2021.

Table 2: Error statistics of the ground control points in both SfM-models

| Model | No. of Control Points (Set 2) | RMSE Control Points [m] | | | | No. of Check Points (Set 1) | RMSE Check Points [m] | | | | No. of z-Val Points (Set 3) | RMSE [m] |
|---|---|---|---|---|---|---|---|---|---|---|---|---|
| | | X | Y | Z | TOT | | X | Y | Z | TOT | | Z |
| 2013 | 33 | 0.14 | 0.12 | 0.10 | **0.21** | 32 | 0.41 | 0.37 | 0.20 | **0.59** | 9 | 0.37 |
| 2021 | 31 | 0.20 | 0.10 | 0.16 | **0.28** | 36 | 0.21 | 0.10 | 0.18 | **0.30** | 11 | 0.12 |

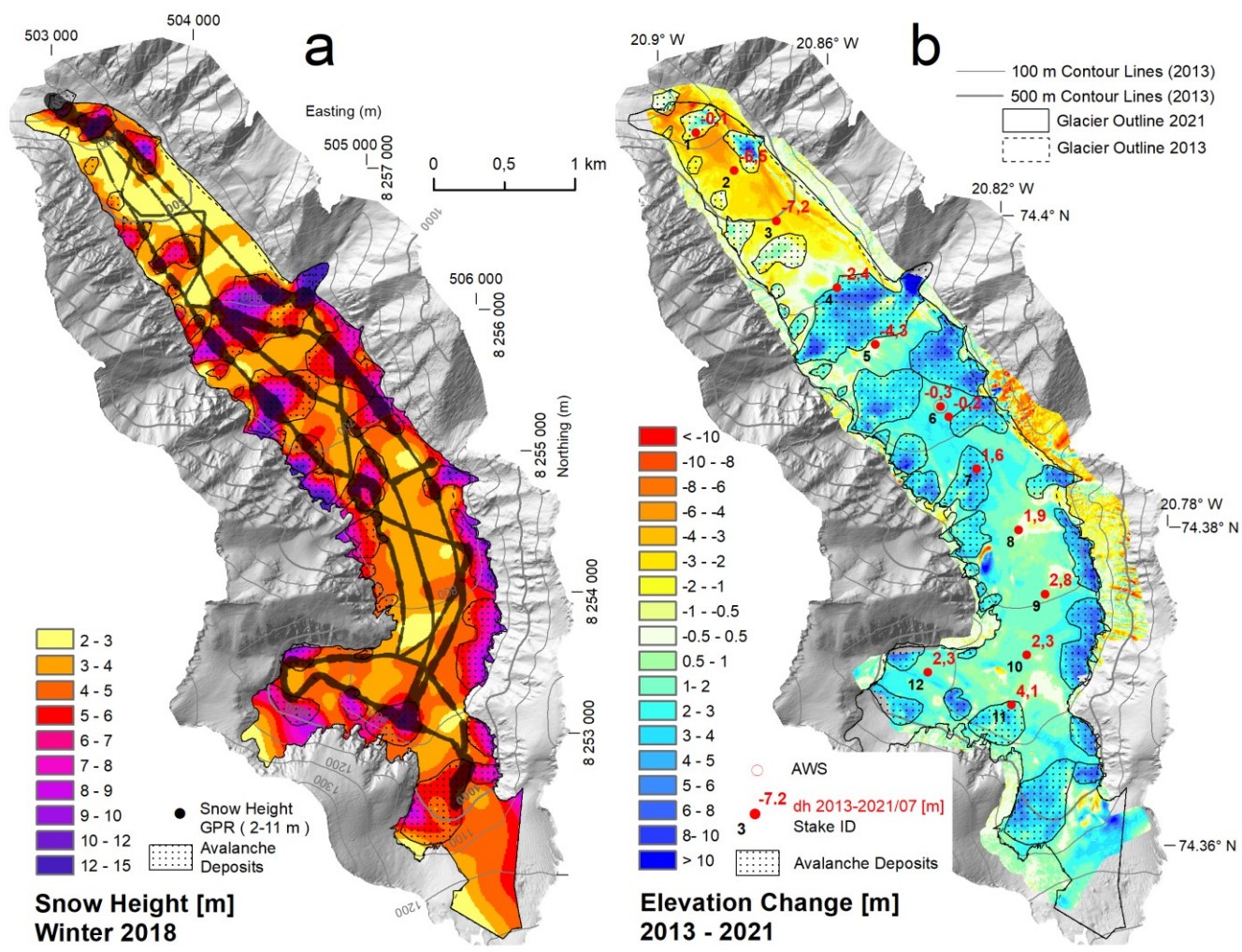

**Figure 6: a) Measured (GPR) and extrapolated snow heights in winter 2018 and delineation of avalanche affected areas. b) Elevation change between 18.8.2013 and 27.7.2021. Cumulative ablation at the stakes for the same period are shown in red (in m).**

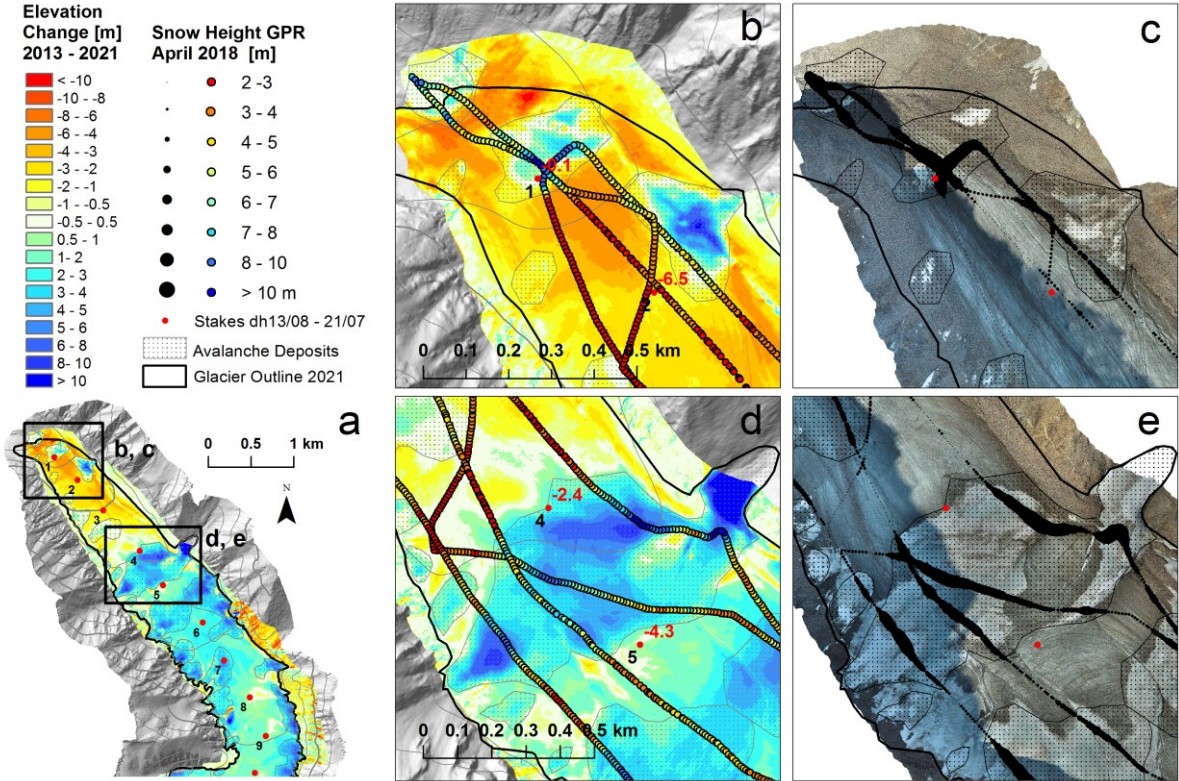

Figure 7: a) Overview and (b, d) close-ups of Elevation Changes and (c, e) Ortho**photo 2021 together with GPR snow height data of spring 2018 and measured ablation at the stakes (in m).**

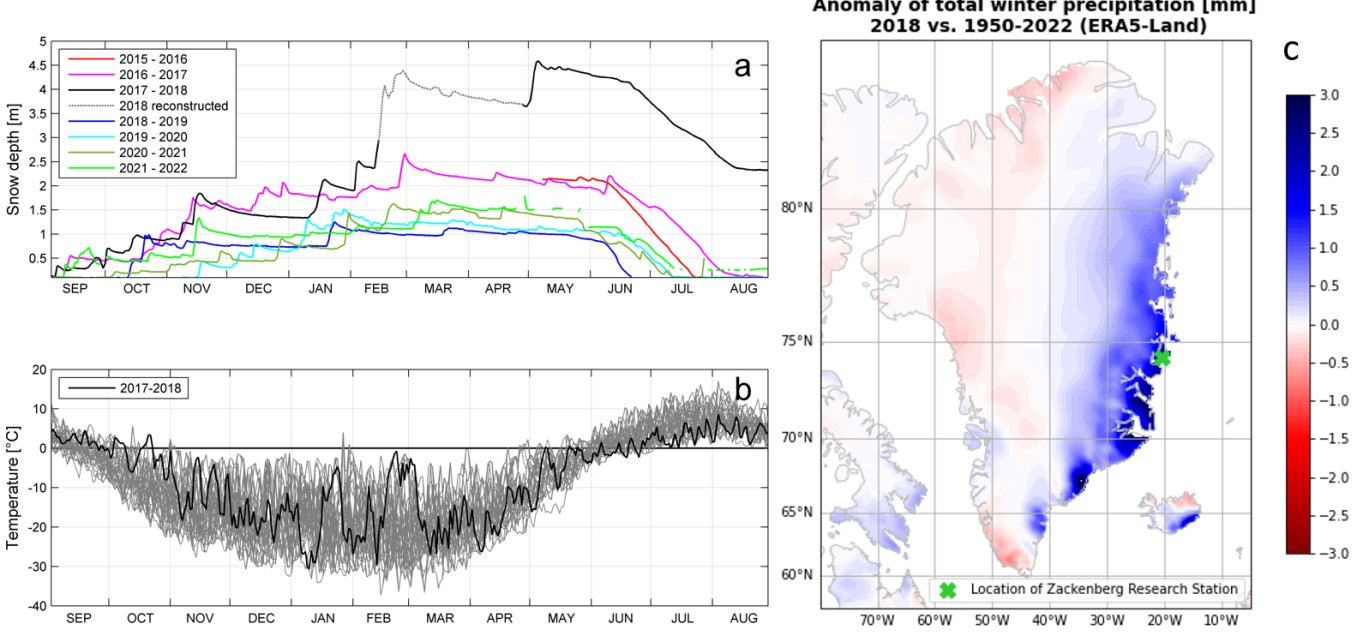

**Figure 8: a) Continuous snow depth record from the AWS on Freya Glacier (680 m a.s.l.) since May 2016. b) Daily mean temperature at Zackenberg (37 m a.s.l.) c) Anomaly of ERA 5 cumulative precipitation (SEP-MAY) of 2018.**

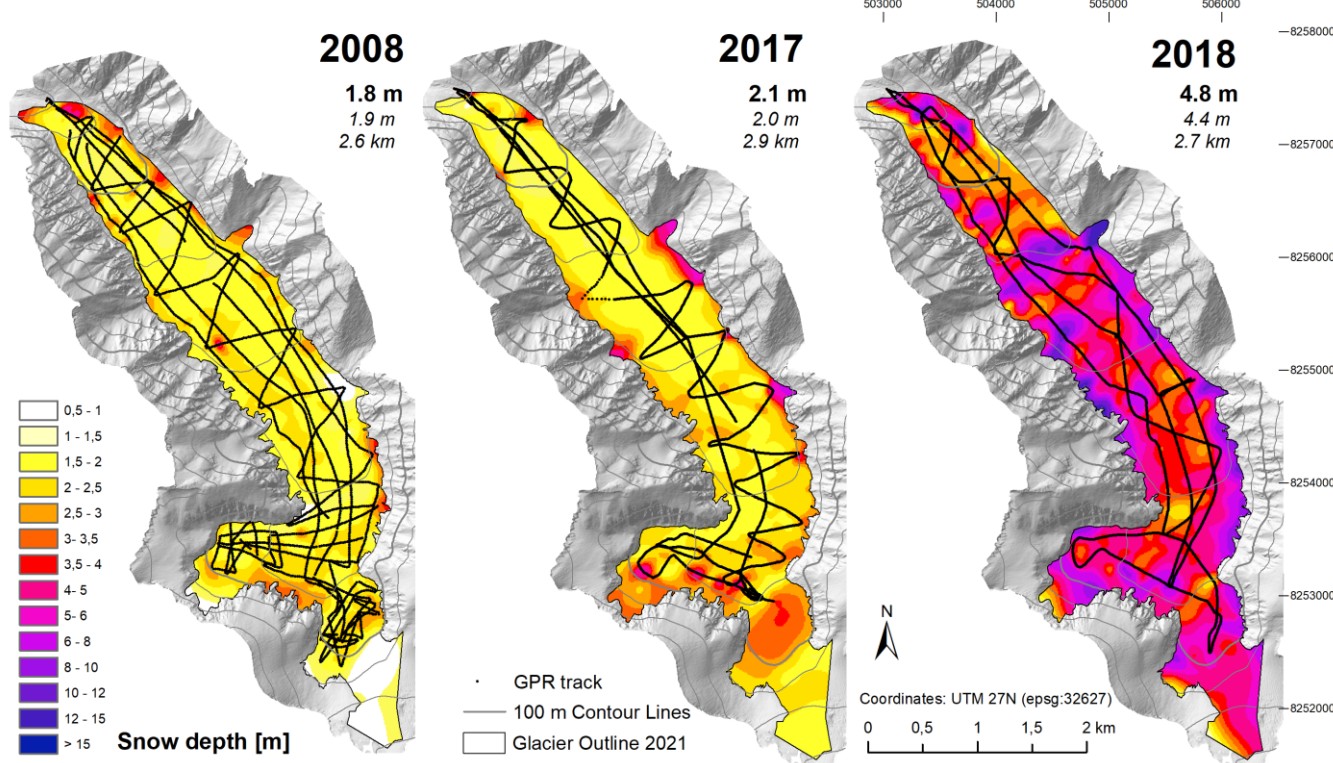

**Figure 9: End of winter snow depth maps in years with a detailed GPR survey. Mean snow depth of the interpolated grid is given in bolt, arithmetic mean of the individual GPR snow depth points is given in italic. Length of the GPR track is given in km.**

**Table 3: Spatial mean values of the winter balance 2018 and the multi-year geodetic mass balance.**

|  | Spatial Mean on Total Glacier Area | Spatial Mean on Glacier Area affected by avalanches 2018 | Spatial Mean on Glacier Area NOT affected by avalanches 2018 |
|---|---|---|---|
| Surface Area 2021 [km²] | 5.54 | 1.98 | 3.55 |
| Surface Area [%] | 100% | 36% | 64% |
| Elevation change [m] 08/2013 - 07/2021 | 1.56 +/- 0.15 | 3.18 | 0.67 |
| Geodetic mass balance [m w.e.] 08/2013 - 07/2021 | 1.33 +/- 0.22 |  |  |
| Winter 2018 snow height [m] | 4.8 | 6.2 | 4.0 |
| Winter mass balance [m w.e.] (constant density) | 1.85 | 2.40 | 1.54 |
| Winter mass balance [m w.e.] (5% density increase) | 1.89 | 2.52 | 1.54 |
| Winter mass balance [m w.e.] (10% density increase) | 1.93 | 2.64 | 1.54 |

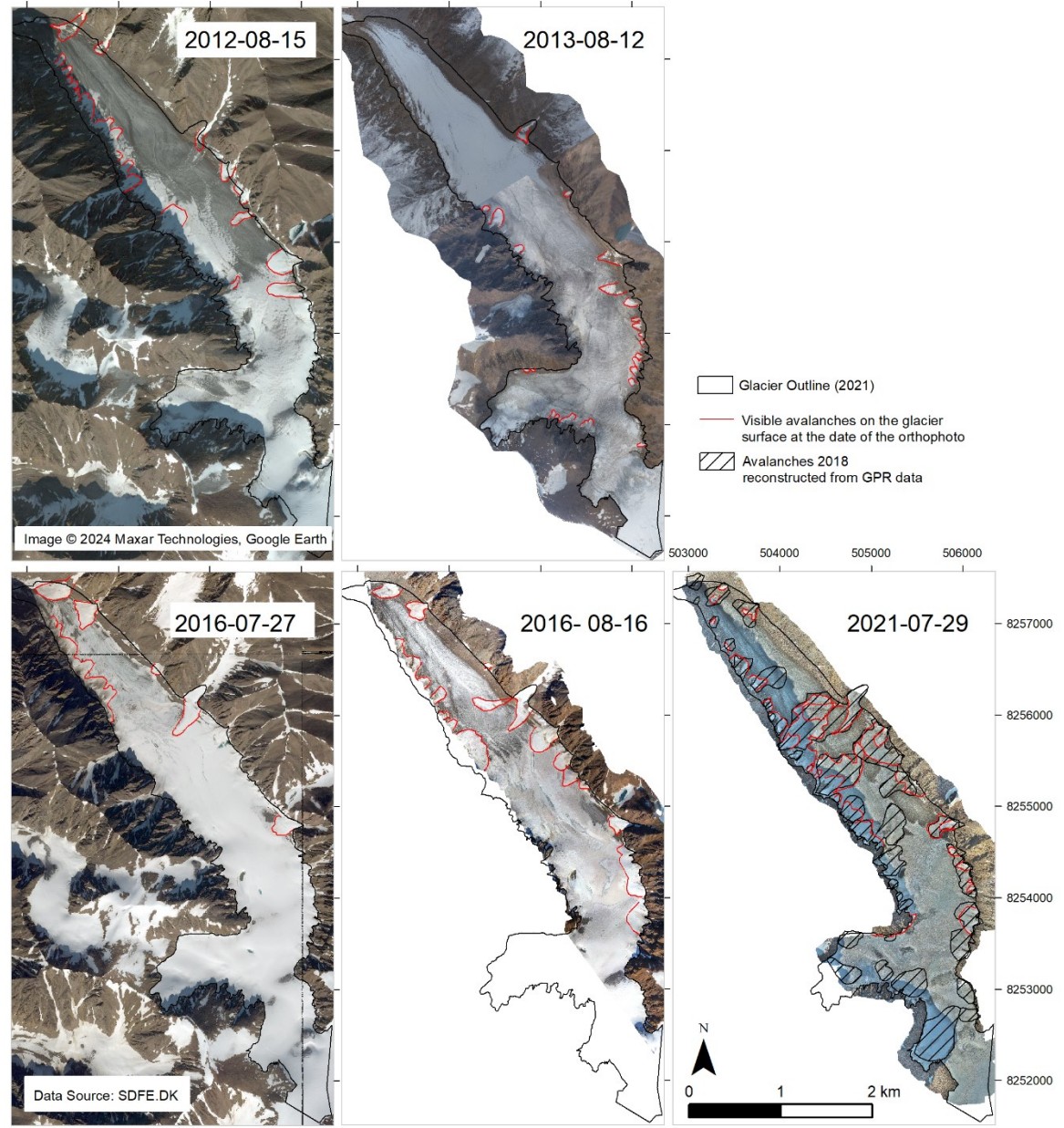

**Figure 10: Visible avalanche deposits in high resolution orthophotos of Freya Glacier of 2012, 2013, 2016 and 2021.**

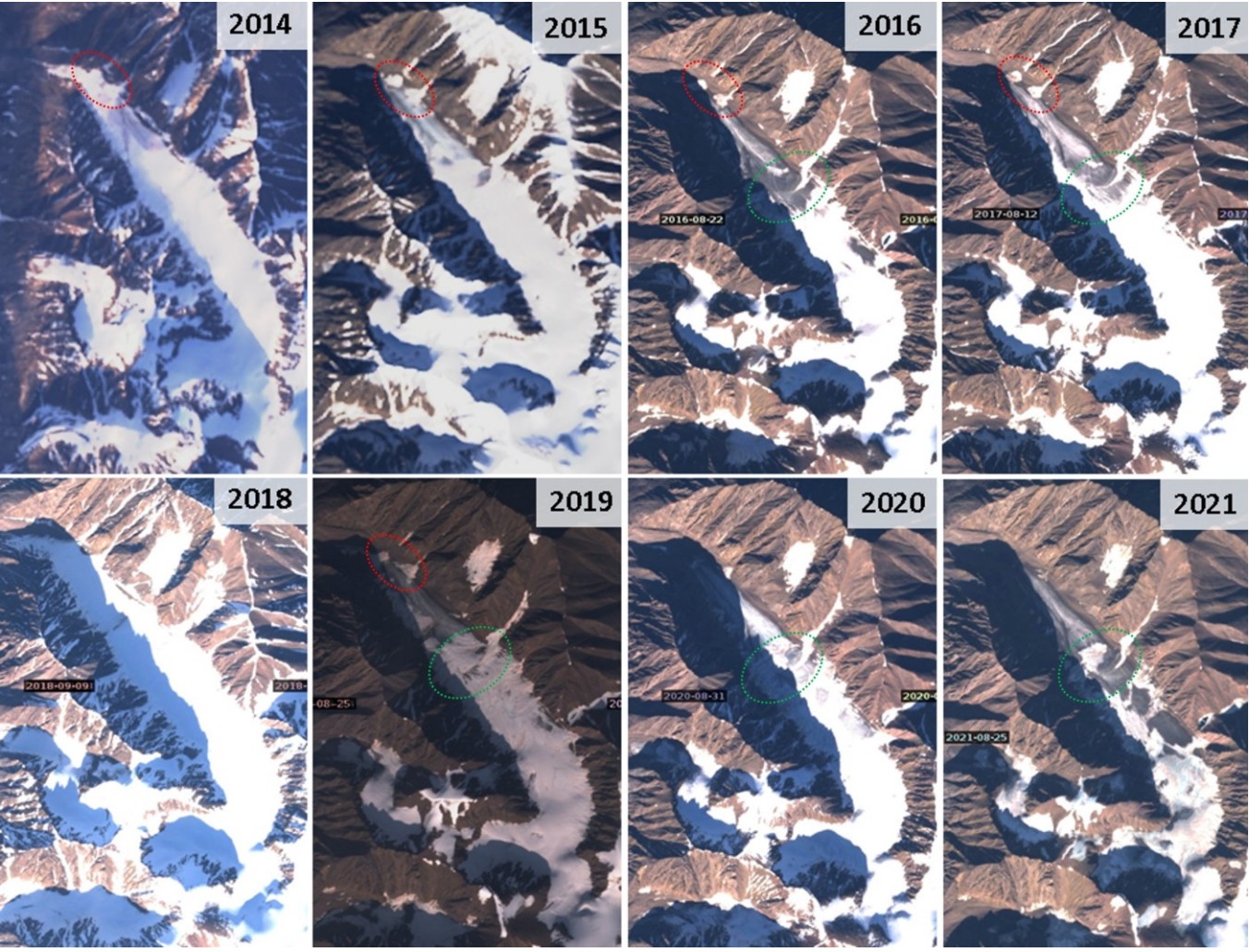

550

**Figure 11: Landsat (2014-2015) and Sentinel 2 (2016-2021) images show the snow cover extent at the end of the ablation season in the relevant mass balance period. Avalanche affected areas are visible in the ablation zone (red) and in some years also in the middle of the glacier (green), where large side valleys are located. Landsat images 2014 and 2015 courtesy of the U.S. Geological Survey. Copernicus Sentinel 2 data 2016-2021, processed by ESA, were retrieved from Sentinel Hub.**

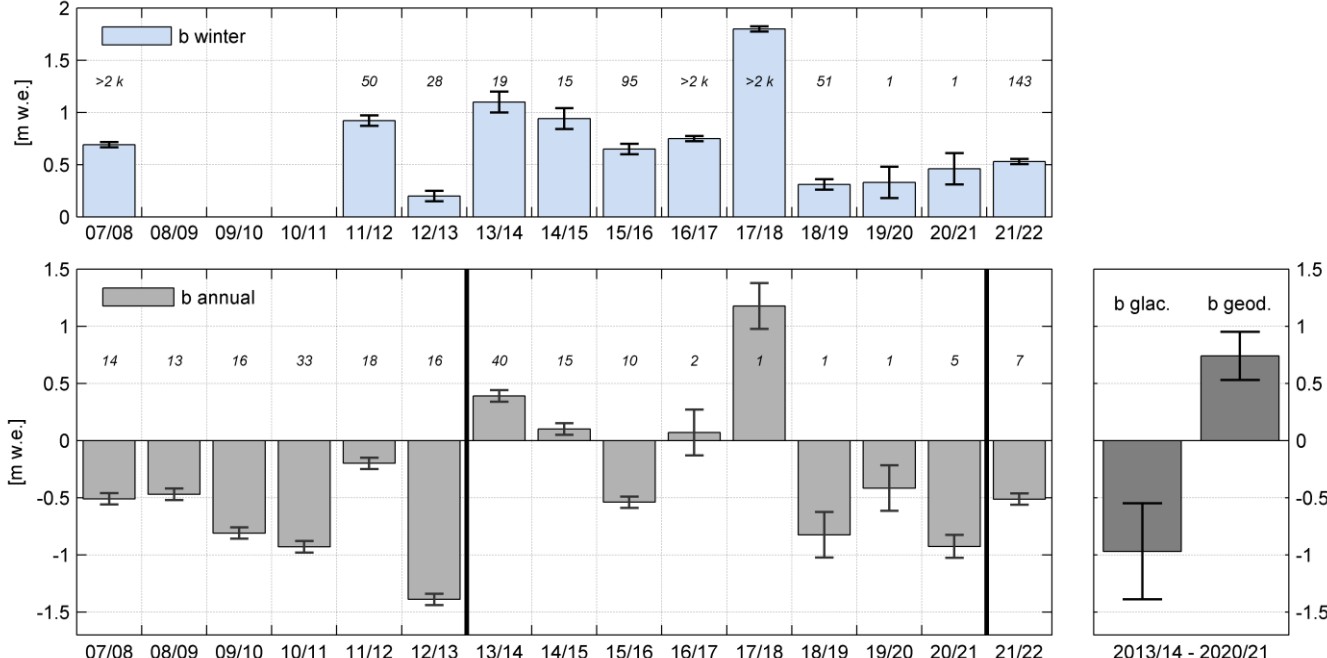

**Figure 12: Left panels: Time series of specific winter mass balances (top), and specific annual mass balances (bottom) with their estimated uncertainties. The number of point observations available for the mass balance calculation of individual years (winters) is shown as italic numbers. E. g. winter mass balance 2017/18 is based on more than 2000 point, while annual balance 2017/18 is based on one point observation only. The high number of point observations in winter 2017/18 corresponds to 10 m along track mean snow depth values of the extensive GPR survey. Right panel: Comparison of the cumulative glaciological and geodetic mass balance 2013/14 – 2020/21 and their related uncertainties.**

---

[i] According to the Language Secretariat of Greenland (Oqaasileriffik.gl) the official name is spelled as *Frejagletsjer* (formerly *Frejagletcher*). While (Ahlmann, 1946) used *Fröya Glacier*, in (Higgins, 2010) the glacier was also spelled as *Fröjabreen, Frøya Glacier* and *Fröya Glacier*. In recent scientific literature (Schöner et al., 2009) the spelling *Freya Glacier* has been used.