# Peer review of "Accumulation by avalanches as significant contributor to the mass balance of a peripheral glacier of Greenland"

_The Cryosphere, 2023_

## Referee Comment (RC2)

Review of Hynek et al.: "Accumulation by avalanches as significant contributor to the mass balance of a High Arctic mountain Glacier"

In this paper, Hynek and colleagues present and analysis of very interesting data collected on Freya Glacier, one of Greenland's peripheral glaciers. They report annual glacier-wide mass balance observations from the glaciological method (2007 to 2022), as well as a geodetic survey between summer 2013 and summer 2021. They find close to equilibrium mass balance conditions, with a geodetic mass balance of 0.25 +/- 0.21 m w.e. over the eight years of survey. They link the observed pattern of elevation changes with the imprints of large avalanches that affected Freya Glacier in winter 2017/18, and that were investigated with an extensive ground penetrating radar survey.

The study is very interesting, and the data collected are of remarkable quality. This contribution is a long awaited one, as the topic of avalanche contribution to glacier mass balance remains poorly explored. Still I recommend major revisions, because there are two main points that would require some attention.

Major comments:

   1- Quantification of the avalanche contribution in the geodetic mass balance

Here I am sorry to be direct, and I might be wrong, but I am not sure that the method presented by the authors to separate the geodetic mass balance into areas that are affected by avalanches and areas not affected by avalanches is actually valid (L161-164). I do not understand why the mean elevation change of an areas that were not mapped as avalanche deposits in winter 2017/18 should not be affected by avalanches as well. If we write the kinematic relation for surface elevation changes, we get:

$$\frac{\partial h}{\partial t} = \frac{\dot{b}_S}{\rho} + w_S - u_S \frac{\partial h}{\partial x} - v_S \frac{\partial h}{\partial y}$$

With h being the glacier surface, $\frac{\dot{b}_S}{\rho}$ the surface mass balance normalized by density, $w_S$ the vertical velocity, $u_S$ and $v_S$ the components of the horizontal velocity and $\frac{\partial h}{\partial x}$ and $\frac{\partial h}{\partial x}$ the components of the local slope. $w_S - u_S \frac{\partial h}{\partial x} - v_S \frac{\partial h}{\partial y}$ is named the emergence velocity and $-u_S \frac{\partial h}{\partial x} - v_S \frac{\partial h}{\partial y}$ is named the advection of topography. This equation tells that the elevation change is the sum of surface mass balance and emergence velocity (or divergence of ice flux). The integral of the elevation change is equal to the integral of the mass balance only if done on a closed surface, which is not the case here, as there is a spatial continuity between the areas affected by avalanches and the areas not affected. As a simple example, one can imagine the deposit from an avalanche that would be advected by the flow and could change location within an elevation band, or change elevation band. Avalanches have also likely a non-local influence on the emergence velocity, simply because they lead to larger mass inputs, and thus larger ice flow. One solution to circumvent this difficulty is to calculate the distributed surface mass balance of the glacier from the elevation change map (e.g., Van Tricht et al., 2021; Vincent et al., 2021), but this required accurate knowledge of the glacier surface velocity, thickness, and to a lesser extent thermal regime.

I might also be wrong in my reasoning, and I think that the authors are absolutely right in their interpretation of the large impact of the winter 2017/18 avalanches, I would just be more careful on the quantitative side. Qualitative arguments are already quite strong regarding the persistence of snow three years after the event, and the good match between positive elevation changes and location of the deposits.

2- How frequent are the avalanches/how exceptional is winter 2017/18?

While the authors demonstrate clearly that the winter 2017/18 corresponds to a mass balance that is two sigma above the average and report that they are not aware of other large avalanches that affected Freya Glacier, I am not convinced that the glacier is not avalanche prone on "normal" years for some of its areas. In the hillshade from August 2013, there are signs of avalanche deposits or cones on the glacier surface, especially at the foot of the north east face, but also on the topographic right, around 600 m a.s.l. The authors could discuss whether the winter 2017/18 was exceptional compared with "normal" winters. One option would be to show other snow height maps to highlight the abnormal avalanche deposits. You could also investigate the climate records/reanalysis to assess the causes of this exceptional avalanche activity.

Specific comments:

L30-31: this sentence is not really clear to me. Do you suspect a bias in the data? Or do you observe a shift in the mass balance?

There are limited links between the different paragraphs of the introduction. I think it should be possible to improve it a bit.

L72-73: the reference is an abstract from EGU. Consider removing it?

There are many acronyms in the text. Consider spelling out Freya Glacier instead of its acronym. Same for the MGIC.

Supplement: I found the supplementary material by accident because it is not referred to in the text. I think it is important material, that demonstrates the very high quality of the two photogrammetric surveys, and it should be better emphasized (in L148 for example).

L112: I enjoyed very much looking at the automatic camera photographs! Thanks!

L134, 151-152 and 210-213: the correction applied to the geodetic survey is confusing because it is mentioned are three distinct locations, and inconsistent in some places (typo on the units on L212). I suggest to write from the beginning state that you apply a -0.60 m w.e. correction to the glacier wide mass balance, and potentially introduce the notations you use later on.

L143-144: how are the two DEMs/orthos merged? Consider providing more details about the elevation different on areas that are covered by both surveys.

L154: "If feasible" suggests that you collected other GPR surveys of the snow thickness. It might be interesting to show some results from these surveys to highlight how winter 2017/18 is different from "average" winters.

L156-158: more details are needed about the avalanche deposit delineation. Which criteria do you use?

L158 [IMPORTANT]: what is the impact of this spline interpolation on the average snow thickness? On figure 8, it seems that the maximum snow thickness is not directly observed but extrapolated from the spline function. The pattern looks reasonable to me, as we expect maximum snow thickness close to the edges, but I think some lines about the uncertainty of this interpolation are needed.

L161: what is the value of the "bulk snow density"? Do you have multiple snow density estimates? Do you have density estimates of the avalanche deposits?

L174-179: much more details are needed. First of all, it is not that usual to do fieldwork in spring to calculate annual mass balance. I imagine that there are some logistical constraints that explain this. You need to better explain how you find the ice surface and/or the horizon of the previous year. You also need to provide more details about the calculation of glacier-wide mass balance when only one or two stakes are found. The "statistical relationship" needs to be described, as well as the associate uncertainties.

L230: the current units for the stake measurements are m. This is a bit confusing and it would be better to use m w.e., as we are talking about surface mass balance here. The period is needed as well. On figure 5, the same comment applies: at stake location, the numbers correspond to elevation change (as suggested by the legend), or do they correspond to surface mass balance (as suggested by the text)? You could consider comparing the surface mass balance and elevation change at the stake location, this would give and idea of the impact of the dynamic.

L234: see my major comment 1, I doubt that the method can "predict" the glacier-wide mass balance without avalanches

L243-244: repetition of L230

L245: I find the unit m w.e. a-1 clearer that the unit ma-1 w.e. that is used here. Consider changing.

Discussion: the transition from the result section to the discussion is rather abrupt. Consider adding a few sentences to make a more seamless transition.

L249-262: this discussion is very interesting, but it could be expanded a bit by testing the impact of the different choices of density on the results?

L257: issues with the citation formatting

In general, the discussion could be sharpened and expended a bit. One aspect could be the climate context of Freya Glacier. I assume that there are very few climate record in the area, but it would be interesting to see whether the winter 2017/18 stands out in the climate record as particularly wet, and then cold or warm.

L297-299: I agree with this statement, but it is never mentioned in the text before so it is a bit surprising to find it in the conclusion.

The data availability statement could be more precise. The mass balance data are available through WGMS I assume? The DEMs or dh maps and snow depth maps could potentially be deposited on a repository.

References

Van Tricht, L., Huybrechts, P., Van Breedam, J., Vanhulle, A., Van Oost, K., and Zekollari, H.: Estimating surface mass balance patterns from unoccupied aerial vehicle measurements in the ablation area of the Morteratsch–Pers glacier complex (Switzerland), The Cryosphere, 15, 4445–4464, https://doi.org/10.5194/tc-15-4445-2021, 2021.

Vincent, C., Cusicanqui, D., Jourdain, B., Laarman, O., Six, D., Gilbert, A., Walpersdorf, A., Rabatel, A., Piard, L., Gimbert, F., Gagliardini, O., Peyaud, V., Arnaud, L., Thibert, E., Brun, F., and Nanni, U.: Geodetic point surface mass balances: a new approach to determine point surface mass balances on

glaciers from remote sensing measurements, The Cryosphere, 15, 1259–1276,
https://doi.org/10.5194/tc-15-1259-2021, 2021.

---

## Author Comment (AC1)

**Response to Review 1 of "Accumulation by avalanches as significant contributor to the mass balance of a High Arctic mountain Glacier"**

Bernhard Hynek, Daniel Binder, Michele Citterio, Signe Hillerup Larsen, Jakob Abermann, Geert Verhoeven, Elke Ludewig, Wolfgang Schöner. January 2024

RC: Reviewer Comment  AR: Author Response

This is an interesting paper reporting the effect of an exceptional avalanche cycle in 2018 on the mass balance of a small polythermal glaciers in NE Greenland. The analysis is based on an extensive data set of glaciological mass-balance measurements and two DEMs of the glacier. The underlying assumptions for volume-to-mass conversion that are used in the paper to compute geodetic mass balance need to be improved.

We appreciate the reviewer's thorough and insightful review of our manuscript! In the following, we respond to the individual comments and describe how we would like to address them in a revised version of the manuscript.

Comments:

The introduction states that mountain glaciers and ice caps are responsible for ~8% of the world's land ice contribution to sea-level rise during the last 60 years. This seems implausibly low. Please check the papers quoted in line 46 and IPCC reports to verify this.

This number refers to the glaciers and ice caps in Greenland, not to glaciers and ice caps globally. As that was obviously unclear, we will clarify that in the text.

My most important comment concerns the methodology to convert elevation change to geodetic mass balance, which contains a possible error that has to do with the effect of ice flow and densification. The use of surface densities for the volume-to-mass conversion (described in the paragraph in lines 146 to 152), neglects the effect of the conversion of firn to ice at depth and the related effect of submergence/emergence velocity in the accumulation and ablation areas, see Huss (2013). The analysis of this problem by Huss (2013) is referenced on page 9 in the paper and in the discussion section but Huss' conclusion that an average conversion factor close to ice-density (850±60 kg/m³) is often appropriate for several-years-long periods or longer is not properly used in my opinion. The snow avalanches that the paper concludes led to (part of) the elevation increase in the accumulation area fell in the spring of 2018 and their deposits are, therefore, four years old in late summer/fall 2021 (in terms of the number of summers they have "experienced"), when the second DEM was measured. Densification due to continued snow/firn metamorphosis in the second to fourth year after deposition may be expected to have taken place and increased the density of the buried avalanche deposits. The density of the buried snow avalanche deposits in 2021 must, therefore, be substantially larger than typical surface density in the fall (600 kg/m³). In addition, part of the thickening in the accumulation area of Freya Glacier in 2013–2021 may have been do to "... continued thinning in lower elevations and thickening in higher elevations", which has been observed at many glaciers in Northeast Greenland (and elsewhere such as in Iceland) in recent years as mentioned in lines 50–54 of the paper. Geometry and volume changes due to such prolonged adjustment of glaciers to changes in mass balance must be expected to be captured with a volume-to-mass conversion factor close to the value recommended by Huss (2013). The authors should discuss this problem with reference to Huss (2013) and perhaps adopt some appropriate value, higher than 600 kg/m³, for an estimate of the density of the remaining avalanche deposits in the accumulation area but adopt a conversion factor close to Huss' recommendation for other volume changes during the period 2013–2021 that may have taken place). This may be difficult to differentiate but should at least be discussed. If there is some knowledge of density profiles at depth for Freya Glacier, or if observations at other polythermal glaciers under similar conditions are available, density values for four-years-old firn might be appropriate for the buried

avalanche deposits. If such observations indicate density > ~(750–800) kg/m³ for several-years-old firn at the expected depth of the buried avalanche deposits on Freya Glacier in 2021, using Huss' recommended value for the entire volume change integrated over the entire glacier may perhaps be the simplest and best choice (?).

We are grateful for this important comment! The reviewer calls our attention to a methodological error that we have committed in the volume to mass conversion by assigning surface densities to elevation changes on different areas of the glacier. We will build on this, discuss the subject in more detail and integrate the density estimation over the entire surface and use the scalar result to convert volume to mass. As there are only a few observations on snow and firn density available for Freya Glacier, we will use density profiles from the literature, as the reviewer suggested.

The easiest way to see the problem with using local surface densities to convert elevation changes to geodetic mass balance is to imagine a surface lowering in the accumulation area due to an ice-flow perturbation that is exactly compensated with an equal surface height increase in the ablation area. The use of surface densities leads to a prediction of a considerable mass increase in this case but it is obvious that the mass change is in fact zero.

Thanks for clarifying this; we agree!

The arguments of the authors for using firn density of 600 kg/m³ for the avalanche deposits (and other volume changes due to an elevation increase) comes first in the discussion section. Part of this discussion should be presented already in the methods section as this is the basis for the rest of the paper. Then the discussion might include further elaboration about this question. From the discussion section, it appears that the entire (positive) elevation change in the accumulation area is assumed to have the density (or volume-to-mass conversion factor) of 600 kg/m³ which seems low for other possible contributions of to an elevation increase in the accumulation area, as mentioned above.

Thank you again, we will move the main part of the density assumptions to the methods section.

I find it hard to understand the discussion in the paragraph in lines 274–278 on page 9. It is not clear how the contribution of the avalanches to the winter balance of 2018 is different from the contribution of the avalanches to the mass balance of the period 2013–2021. Of course such a difference can be due to an error, but physically it does not make sense to discuss this as a real quantitative difference. The avalanches are a definite event that deposited a certain amount of snow on the surface of the glacier. It sounds confusing to discuss this contribution to vary with time due to later melting that must be hard to differentiate from melting of other positive contributions to the mass balance of the glacier from 2018 to 2021.

Thank you very much, we agree. This part has been hard to understand and needs rewording, considering your comments. We will change that.

**Minor and editorial comments:**

In figure 5b (and the same figure in the graphical abstract), the legend shows a special pattern to denote avalanche deposits but the map does not seem to show these deposits (the avalanche deposits are shown in figure 5a but not 5b).

line 21: add "°" in "20.82°W"

line 45: perhaps say "their recent contribution to mass loss from Greenland and global sea-level rise is disproportionately"

line 50: perhaps say "has accelerated globally during"

line 59: perhaps say "in Greenland are monitored"

line 62: perhaps say "both at 74°N"

line 113: period missing at end of sentence

line 125: perhaps say "Snowfall on 14th August"

line 144: perhaps say "These parts of the glacier"

line 144: perhaps say "April 2018" to be consistent with line 169?

line 144: perhaps say "total length of"

line 158: perhaps say "onto a grid of"

line 185: perhaps say "poorly covered"

line 189: perhaps say "on the adjacent ridges"

line 191: perhaps say "worse than"

line 201: drop "of the glacier"

line 203: perhaps say "mainly at elevations"

line 207: perhaps say "large side valleys"

line 207: perhaps say "for the entire glacier" rather than "for the total glacier area"

line 236: perhaps say "larger than the lower bound"

line 245: perhaps say "The bias with respect to"

Thanks for your editorial suggestions. We will include those in the revised submission!

Excessive use of acronyms make the text awkward to read in places, especially because the paper is otherwise generally well written. It sounds awkward to use the acronym "FG" about the Freya Glacier, which is the main subject of the paper with a relatively short name that deserves to be written out in full when this glacier is mentioned. In some places, the full name can be written as just "glacier" or "the glacier", when the context is clear, so the use of the full name will not make the text much longer. "FG" is used 12 times in the manuscript, sometimes up to three times in the same paragraph. The acronym "MGIC" for "mountain glaciers and ice caps" is also awkward and used much too often. The paragraph in lines 56 to 60 would, for example, be much easier to read without this uncommon acronym. Try to use as few acronyms as possible. In many cases, a minor rewording will eliminate the acronym and make the text flow better.

Thank you for this comment. We will minimize acronyms in the text. We used the acronym MGIC for mountain glaciers and ice caps of Greenland, to distinguish them from the Greenland ice sheet. We will use peripheral glaciers (of Greenland) instead, where we think it is necessary.

The use of hyphens ("-"), en-dashes ("–") and minus signs ("−") in composite words, negative numbers, number ranges and date ranges is inconsistent in many places. Use an en-dash or a proper minus sign for all negative numbers, also in superscripts such as "a$^{-1}$", and for all number and date ranges. Since you write "high-resolution DEM", you should probably also write "sea-level rise", and similarly for other compound adjectives (very many instances). The unit "meters water equivalent per year" should be written "m w.e. a$^{-1}$", not "m a$^{-1}$ w.e. "

Thank you for the useful remark, we will change that accordingly.

---

## Author Comment (AC2)

**Response to Review 2 of "Accumulation by avalanches as significant contributor to the mass balance of a High Arctic mountain Glacier"**

Bernhard Hynek, Daniel Binder, Michele Citterio, Signe Hillerup Larsen, Jakob Abermann, Geert Verhoeven, Elke Ludewig, Wolfgang Schöner. January 2024

RC: Reviewer Comment          AR: Author Response

In this paper, Hynek and colleagues present and analysis of very interesting data collected on Freya Glacier, one of Greenland's peripheral glaciers. They report annual glacier-wide mass balance observations from the glaciological method (2007 to 2022), as well as a geodetic survey between summer 2013 and summer 2021. They find close to equilibrium mass balance conditions, with a geodetic mass balance of 0.25 +/- 0.21 m w.e. over the eight years of survey. They link the observed pattern of elevation changes with the imprints of large avalanches that affected Freya Glacier in winter 2017/18, and that were investigated with an extensive ground penetrating radar survey.

The study is very interesting, and the data collected are of remarkable quality. This contribution is a long awaited one, as the topic of avalanche contribution to glacier mass balance remains poorly explored. Still I recommend major revisions, because there are two main points that would require some attention.

We appreciate the reviewer's thorough and insightful assessment of our work! In the following, we respond to the individual comments and describe how we plan to address them in a revised version of the manuscript.

**Major comments:**

1- Quantification of the avalanche contribution in the geodetic mass balance

Here I am sorry to be direct, and I might be wrong, but I am not sure that the method presented by the authors to separate the geodetic mass balance into areas that are affected by avalanches and areas not affected by avalanches is actually valid (L161-164). I do not understand why the mean elevation change of an areas that were not mapped as avalanche deposits in winter 2017/18 should not be affected by avalanches as well. If we write the kinematic relation for surface elevation changes, we get:

$$\partial h \partial t = bS\rho + wS - uS\partial h\partial x - vS\partial h\partial y$$

With h being the glacier surface, $bS\rho$ the surface mass balance normalized by density, $wS$ the vertical velocity, $uS$ and $vS$ the components of the horizontal velocity and $\partial h\partial x$ and $\partial h\partial x$ the components of the local slope. $wS - uS\partial h\partial x - vS\partial h\partial y$ is named the emergence velocity and $-uS\partial h\partial x - vS\partial h\partial y$ is named the advection of topography. This equation tells that the elevation change is the sum of surface mass balance and emergence velocity (or divergence of ice flux). The integral of the elevation change is equal to the integral of the mass balance only if done on a closed surface, which is not the case here, as there is a spatial continuity between the areas affected by avalanches and the areas not affected. As a simple example, one can imagine the deposit from an avalanche that would be advected by the

flow and could change location within an elevation band, or change elevation band. Avalanches have also likely a non-local influence on the emergence velocity, simply because they lead to larger mass inputs, and thus larger ice flow. One solution to circumvent this difficulty is to calculate the distributed surface mass balance of the glacier from the elevation change map (e.g., Van Tricht et al., 2021; Vincent et al., 2021), but this required accurate knowledge of the glacier surface velocity, thickness, and to a lesser extent thermal regime.

I might also be wrong in my reasoning, and I think that the authors are absolutely right in their interpretation of the large impact of the winter 2017/18 avalanches, I would just be more careful on the quantitative side. Qualitative arguments are already quite strong regarding the persistence of snow three years after the event, and the good match between positive elevation changes and location of the deposits.

Thank you for this very relevant comment on the influence of glacier dynamics, namely the emergence velocity, which has the potential to alter the results of our quantitative assessment. We missed to tackle this question in depth in our initial submission. We also appreciate your suggestions for solving this problem based on the recent work on the subject.

Freya Glacier is a polythermal glacier with temperate ice only in a limited area near to the bottom of the glacier. Surface slope is generally rather small (0-20°) over large areas of the glacier. Observed horizontal surface velocities at the stakes are on average 6 m a$^{-1}$ and computed vertical velocities (derived from measurements of ablation and surface elevation) are between -0.6 and 0.6 m a$^{-1}$. For the 8 year period 2013 - 2021, for which the elevation change was measured, glacier motion roughly adds a mean horizontal displacement of ~ 50 m and emergence/submergence between about -5 m and +5 m.

The slow glacier dynamics tempted us to neglect the influence of emergence velocity in our quantitative assessment without discussing it in the manuscript. In other words, we based our analysis on the following assumptions:

1. Although the surfaces over which we integrate the elevation changes are not closed, as the reviewer correctly notes, we assume that the emergency velocity over these areas is negligible compared to the mean elevation changes.
2. The distribution of emergence velocity on the glacier is not significantly changed by the avalanche deposits (and/or this effect would be hard or impossible to quantify) or the related change in emergence velocity is significantly smaller than the elevation change by the avalanche deposits.

We plan to address this question by clearly describing and discussing the assumptions on which our assessment is based and discuss the introduced uncertainty of these assumptions. We note, that other uncertainties or errors also exist, mainly the unknown density of the avalanche deposits or the uncertainty of the delineation of avalanche deposits, so the quantification of the mass contribution by avalanches should be seen as a rough estimate.

2- How frequent are the avalanches/how exceptional is winter 2017/18?

While the authors demonstrate clearly that the winter 2017/18 corresponds to a mass balance that is two sigma above the average and report that they are not aware of other large avalanches that affected Freya Glacier, I am not convinced that the glacier is not avalanche prone on "normal" years for some of its areas. In the hillshade from August 2013, there are signs of avalanche deposits or cones

on the glacier surface, especially at the foot of the north east face, but also on the topographic right, around 600 m a.s.l. The authors could discuss whether the winter 2017/18 was exceptional compared with "normal" winters. One option would be to show other snow height maps to highlight the abnormal avalanche deposits. You could also investigate the climate records/reanalysis to assess the causes of this exceptional avalanche activity.

Thank you for this comment, we will add more information/observations on this interesting question. As we wrote in the original submission (L 216), remnants of avalanches have been observed also in other years, but the coverage of the avalanche deposits (and most likely also the mass input) was much smaller than in 2018. To provide more information on this, we will compile all available observations of avalanches from previous years. This includes: Reduced melt rates at stake 1 by avalanches and orthophotos and satellite imagery that show imprints of avalanches in "normal years".

To demonstrate the climatologically exceptional winter of 2018 compared to other winters in the region, we will add winter precipitation data from the station Zackenberg and from reanalysis and contextualize it with the snow depth time series at Freya Glacier since the beginning of the measurements.

**Specific comments:**

L30-31: this sentence is not really clear to me. Do you suspect a bias in the data? Or do you observe a shift in the mass balance?

We observe a bias in the direct mass balance data (see Fig.8), which appears too negative in comparison with the geodetic mass balance 2013-2021. As for now, we do not know the reason for this bias, therefore we recommend a reanalysis of the annual direct mass balances following the procedure of Zemp et al (2013). While in mass balance years before 2016/17 the annual mass balance is based on more than 10 point observations, in the following years the annual mass balance is only based on one point observation (at the AWS) which introduces a much higher uncertainty. Apparently, this phrase causes confusion, so we will describe this more clearly in the revised version of the paper.

Since 2016, mass balance fieldwork usually takes place in spring. During the fieldwork we measure the ablation of the previous year at the stakes. If the snow cover in winter is too thick, the stakes do not stick out of the snow and are hard to find. This is the reason why high accumulation rates led to fewer mass balance point observations in the past.

There are limited links between the different paragraphs of the introduction. I think it should be possible to improve it a bit.

Thanks for this comment. We will insert a few more sentences to make the transitions smoother.

L72-73: the reference is an abstract from EGU. Consider removing it?

Yes, we will remove it.

There are many acronyms in the text. Consider spelling out Freya Glacier instead of its acronym. Same for the MGIC.

Thank you for this comment, this was also mentioned by reviewer 1. We will change this accordingly. Instead of MGIC we will use the term peripheral glaciers (of Greenland).

Supplement: I found the supplementary material by accident because it is not referred to in the text. I think it is important material, that demonstrates the very high quality of the two photogrammetric surveys, and it should be better emphasized (in L148 for example).

Thank you for this comment and the positive words about our data. We will describe the error statistics on stable terrain more prominently in the text and refer to the supplementary material more clearly in the text.

L112: I enjoyed very much looking at the automatic camera photographs! Thanks!

We are pleased to hear that you enjoyed it!

L134, 151-152 and 210-213: the correction applied to the geodetic survey is confusing because it is mentioned are three distinct locations, and inconsistent in some places (typo on the units on L212). I suggest to write from the beginning state that you apply a -0.60 m w.e. correction to the glacier wide mass balance, and potentially introduce the notations you use later on.

Thank you for this comment. We will bring that in line.

L143-144: how are the two DEMs/orthos merged? Consider providing more details about the elevation different on areas that are covered by both surveys.

Thank you for this advice. Yes, there is an area in the middle of the glacier, which is covered by both surveys in 2013. Due to the poor coverage of that area (no near photo points and snow cover in diffuse illumination), surface reconstruction uncertainty in that area is higher than in other parts. We will describe the methodology how we merged the DEMs and we provide a figure of these vertical differences in the supplementary material.

L154: "If feasible" suggests that you collected other GPR surveys of the snow thickness. It might be interesting to show some results from these surveys to highlight how winter 2017/18 is different from "average" winters.

We performed GPR surveys with a similar point observation density in spring of 2008 and 2017, in other years we have fewer snow depth point observations (see point observation number in Figure 8 (top panel). In 2008 and 2017 the mean snow depth was close to average, so that we can calculate an average snow distribution from the two surveys and use this as a basis for determining 2018 anomalies.

L156-158: more details are needed about the avalanche deposit delineation. Which criteria do you use?

The delineation of avalanche deposits is a crucial step in the quantification and we will add more information about it in the revised version of the paper. The delineation of the avalanche affected areas did not follow strict criteria, as this seemed hardly feasible and not beneficial based on the available information. Along the GPR tracks we used a strong increase in GPR snow depth as an

indication for deposits. To complete the delineation in areas without GPR tracks we used a best estimate based on photos of avalanche fracture lines, remnants of avalanches in the orthophoto of 2021 and above average local elevation changes together with likely avalanche flow paths based on topography. Besides better description we will evaluate the impact of delineation uncertainty on the derived contribution of avalanches to the mass balance.

L158 [IMPORTANT]: what is the impact of this spline interpolation on the average snow thickness? On figure 8, it seems that the maximum snow thickness is not directly observed but extrapolated from the spline function. The pattern looks reasonable to me, as we expect maximum snow thickness close to the edges, but I think some lines about the uncertainty of this interpolation are needed.

Thank you for this important comment. We will evaluate and shortly describe the effect of the uncertainty introduced by the snow height interpolation method although we deem this small compared to other error sources.

L161: what is the value of the "bulk snow density"? Do you have multiple snow density estimates? Do you have density estimates of the avalanche deposits?

In spring 2018 we could carry out one snow density measurement within a snow pit next to the AWS (see line 223). We will remove the word "bulk", as it is misleading. Snow density at the snow pit next to the AWS at stake 6 was 385 kg/m³. There were no further snow density measurements possible in 2018. Particularly, there are no observations of the snow density of the avalanche deposits. As the density of snow and firn plays an important role in the calculation of the mass balance and the quantification of the contribution by avalanches, we will add the data of the snow pit to the supplementary material and describe our assumptions on extrapolating the local snow density in more detail.

L174-179: much more details are needed. First of all, it is not that usual to do fieldwork in spring to calculate annual mass balance. I imagine that there are some logistical constraints that explain this. You need to better explain how you find the ice surface and/or the horizon of the previous year. You also need to provide more details about the calculation of glacier-wide mass balance when only one or two stakes are found. The "statistical relationship" needs to be described, as well as the associate uncertainties.

In 2016 the monitoring strategy at Freya Glacier was changed from summer visits to spring visits due to logistical reasons. In summer, Freya Glacier can only be reached by boat and a long walk, while in winter, it is easier and faster accessible on snow scooters, which also allows the transportation of heavier equipment like weather stations or a snow radar. Moreover, one single visit to the glacier per year allows us to determine both the winter and annual balance of the glacier (not in all years). While the ablation stakes can be measured easily, as long as they stick out of the winter snow, it is more difficult to measure accumulation. To get accurate information on the extent of the annual accumulation area, we installed the automatic cameras. We will describe the method of mass balance evaluation and related uncertainties more precisely in the revised version of the paper.

L230: the current units for the stake measurements are m. This is a bit confusing and it would be better to use m w.e., as we are talking about surface mass balance here. The period is needed as well. On figure 5, the same comment applies: at stake location, the numbers correspond to elevation change (as suggested by the legend), or do they correspond to surface mass balance (as suggested by the

text)? You could consider comparing the surface mass balance and elevation change at the stake location, this would give and idea of the impact of the dynamic.

Thank you for this comment. We will clarify this and use only the units m w.e.. In figure 5 we will change the units and clarify that the given value is the local mass balance. In addition we will add a table, where we compare ablation and elevation change at the stakes to show the dynamic component.

L234: see my major comment 1, I doubt that the method can "predict" the glacier-wide mass balance without avalanches

To answer this, please see our answer to your comment 1.

L243-244: repetition of L230

Thank you. We will bring that in line.

L245: I find the unit m w.e. a-1 clearer that the unit ma-1 w.e. that is used here. Consider changing. Discussion: the transition from the result section to the discussion is rather abrupt. Consider adding a few sentences to make a more seamless transition.

Thank you. We will change this accordingly.

L249-262: this discussion is very interesting, but it could be expanded a bit by testing the impact of the different choices of density on the results?

Yes, we will expand the discussion.

L257: issues with the citation formatting

Thanks, we correct that.

In general, the discussion could be sharpened and expended a bit. One aspect could be the climate context of Freya Glacier. I assume that there are very few climate record in the area, but it would be interesting to see whether the winter 2017/18 stands out in the climate record as particularly wet, and then cold or warm.

Thank you for that suggestion. We are happy to add more about the climate context of Freya Glacier and the year of 2018.

L297-299: I agree with this statement, but it is never mentioned in the text before so it is a bit surprising to find it in the conclusion.

Thank you for that comment. We suggest expanding on that question a little more within a new climate context section.

The data availability statement could be more precise. The mass balance data are available through WGMS I assume? The DEMs or dh maps and snow depth maps could potentially be deposited on a repository.

Thank you for that comment. We will add more data and information on data availability here.

References

Van Tricht, L., Huybrechts, P., Van Breedam, J., Vanhulle, A., Van Oost, K., and Zekollari, H.: Estimating surface mass balance patterns from unoccupied aerial vehicle measurements in the ablation area of the Morteratsch–Pers glacier complex (Switzerland), The Cryosphere, 15, 4445–4464, https://doi.org/10.5194/tc-15-4445-2021, 2021.

Vincent, C., Cusicanqui, D., Jourdain, B., Laarman, O., Six, D., Gilbert, A., Walpersdorf, A., Rabatel, A., Piard, L., Gimbert, F., Gagliardini, O., Peyaud, V., Arnaud, L., Thibert, E., Brun, F., and Nanni, U.: Geodetic point surface mass balances: a new approach to determine point surface mass balances on glaciers from remote sensing measurements, The Cryosphere, 15, 1259–1276, https://doi.org/10.5194/tc-15-1259-2021, 2021.

---

## Author Response (AR1)

**Description of Changes to Comments by Reviewer 1**

Bernhard Hynek, Daniel Binder, Michele Citterio, Signe Hillerup Larsen, Jakob Abermann, Geert Verhoeven, Elke Ludewig, Wolfgang Schöner. June 2024

RC: Reviewer Comment  AR: Author Response, description of changes

This is an interesting paper reporting the effect of an exceptional avalanche cycle in 2018 on the mass balance of a small polythermal glaciers in NE Greenland. The analysis is based on an extensive data set of glaciological mass-balance measurements and two DEMs of the glacier. The underlying assumptions for volume-to-mass conversion that are used in the paper to compute geodetic mass balance need to be improved.

We appreciate the reviewer's thorough and insightful review of our manuscript! In the following, we describe how we addressed them in the revised version of the manuscript.

Comments:

The introduction states that mountain glaciers and ice caps are responsible for ~8% of the world's land ice contribution to sea-level rise during the last 60 years. This seems implausibly low. Please check the papers quoted in line 46 and IPCC reports to verify this.

We changed the text to "During the last 60 years mass loss from Greenland's peripheral glaciers comprise ~ 8% of the world's land ice contribution to sea-level rise (Zemp et al., 2019; Frederikse et al., 2020)." to make clear, that the SLR contribution of 8% refers only to Greenland's peripheral glaciers.

My most important comment concerns the methodology to convert elevation change to geodetic mass balance, which contains a possible error that has to do with the effect of ice flow and densification. The use of surface densities for the volume-to-mass conversion (described in the paragraph in lines 146 to 152), neglects the effect of the conversion of firn to ice at depth and the related effect of submergence/emergence velocity in the accumulation and ablation areas, see Huss (2013). The analysis of this problem by Huss (2013) is referenced on page 9 in the paper and in the discussion section but Huss' conclusion that an average conversion factor close to ice-density (850±60 kg/m³) is often appropriate for several-years-long periods or longer is not properly used in my opinion. The snow avalanches that the paper concludes led to (part of) the elevation increase in the accumulation area fell in the spring of 2018 and their deposits are, therefore, four years old in late summer/fall 2021 (in terms of the number of summers they have "experienced"), when the second DEM was measured. Densification due to continued snow/firn metamorphosis in the second to fourth year after deposition may be expected to have taken place and increased the density of the buried avalanche deposits. The density of the buried snow avalanche deposits in 2021 must, therefore, be substantially larger than typical surface density in the fall (600 kg/m³). In addition, part of the thickening in the accumulation area of Freya Glacier in 2013–2021 may have been do to "... continued thinning in lower elevations and thickening in higher elevations", which has been observed at many glaciers in Northeast Greenland (and elsewhere such as in Iceland) in recent years as mentioned in lines 50–54 of the paper. Geometry and volume changes due to such prolonged adjustment of glaciers to changes in mass balance must be expected to be captured with a volume-to-mass conversion factor close to the value recommended by Huss (2013). The authors should discuss this problem with reference to Huss (2013) and perhaps adopt some appropriate value, higher than 600 kg/m³, for an estimate of the density of the remaining avalanche deposits in the accumulation area but adopt a conversion factor close to Huss' recommendation for other volume changes during the period 2013–2021 that may have taken place). This may be difficult to differentiate but should at least be discussed. If there is some knowledge of density profiles at depth for Freya Glacier, or if observations at other polythermal glaciers under similar conditions are available, density values for four-years-old firn might be appropriate for the buried avalanche deposits. If such observations indicate density > ~(750–800) kg/m³ for several-years-old firn at the expected depth of the buried avalanche deposits on Freya Glacier in 2021, using Huss' recommended value for the entire volume change integrated over the entire glacier may perhaps be the simplest and best choice (?).

The easiest way to see the problem with using local surface densities to convert elevation changes to geodetic mass balance is to imagine a surface lowering in the accumulation area due to an ice-flow perturbation that is exactly compensated with an equal surface height increase in the ablation area. The use of surface densities leads to a prediction of a considerable mass increase in this case but it is obvious that the mass change is in fact zero.

The arguments of the authors for using firn density of 600 kg/m³ for the avalanche deposits (and other volume changes due to an elevation increase) comes first in the discussion section. Part of this discussion should be presented already in the methods section as this is the basis for the rest of the paper. Then the discussion might include further elaboration about this question. From the discussion section, it appears that the entire (positive) elevation change in the accumulation area is assumed to have the density (or volume-to-mass conversion factor) of 600 kg/m³ which seems low for other possible contributions of to an elevation increase in the accumulation area, as mentioned above.

We are greatful for this important comment! The reviewer calls our attention to a methodological error that we have commited in the volume to mass conversion by assigning surface densities to elevation changes on different surface classes. We came to the conclusion, that it might be the best choice to use the proposed volume-to-mass conversion factor of 850 + 60 kgm-³, as recommended by Huss (2013) for periods longer than 5 years. Up to date there are no firn density observations available from Freya Glacier.

In addition, we noticed a computation error, that we have commited in averaging surface densities on different areas: Inititally, we converted the average elevation change of 1.56 m into a geodetic mass balance of 0.85 m w.e. This would require a mean density of 545 kgm-³, which is off course far to low. This computation error combined with the new density assumption of 850 + 60 kgm-³ changes the result of the geodetic mass balance from 0.85 m w.e. to 1.33 m w.e. As a consequence, the difference between geodetic and glaciological mass balance is even larger.

I find it hard to understand the discussion in the paragraph in lines 274–278 on page 9. It is not clear how the contribution of the avalanches to the winter balance of 2018 is different from the contribution of the avalanches to the mass balance of the period 2013–2021. Of course such a difference can be due to an error, but physically it does not make sense to discuss this as a real quantitative difference. The avalanches are a definite event that deposited a certain amount of snow on the surface of the glacier. It sounds confusing to discuss this contribution to vary with time due to later melting that must be hard to differentiate from melting of other positive contributions to the mass balance of the glacier from 2018 to 2021.

We deleted these lines, as we made a major changes in the discussion section.

**Minor and editorial comments:**

In figure 5b (and the same figure in the graphical abstract), the legend shows a special pattern to denote avalanche deposits but the map does not seem to show these deposits (the avalanche deposits are shown in figure 5a but not 5b).

We changed the color from the avalanche deposits in figure 5b from light grey into black, in order to improve the visibility.

line 21: add "°" in "20.82°W"  added
line 45: perhaps say "their recent contribution to mass loss from Greenland and global sea-level rise is disproportionately" revised accordingly

line 50: perhaps say "has accelerated globally during" changed to Greenlands peripheral glaciers (to make clear that we are focussing on Greenland here already)

line 59: perhaps say "in Greenland are monitored" revised accordingly

line 62: perhaps say "both at 74°N" revised accordingly

line 113: period missing at end of sentence revised accordingly

line 125: perhaps say "Snowfall on 14th August" revised accordingly

line 144: perhaps say "These parts of the glacier" revised accordingly

line 144: perhaps say "April 2018" to be consistent with line 169? changed accordingly

line 144: perhaps say "total length of" changed accordingly

line 158: perhaps say "onto a grid of" revised accordingly

line 185: perhaps say "poorly covered" revised accordingly

line 189: perhaps say "on the adjacent ridges" we left the wording "glacier and adjacent ridges", as we want to point out that the surface reconstruction extends to the (non glacierized) ridges at both sides of the valley glacier

line 191: perhaps say "worse than" revised accordingly

line 201: drop "of the glacier" revised accordingly

line 203: perhaps say "mainly at elevations" revised accordingly

line 207: perhaps say "large side valleys" revised accordingly

line 207: perhaps say "for the entire glacier" rather than "for the total glacier area" revised accordingly

line 236: perhaps say "larger then the lower bound" revised accordingly

line 245: perhaps say "The bias with respect to" revised accordingly

Excessive use of acronyms make the text awkward to read in places, especially because the paper is otherwise generally well written. It sounds awkward to use the acronym "FG" about the Freya Glacier, which is the main subject of the paper with a relatively short name that deserves to be written out in full when this glacier is mentioned. In some places, the full name can be written as just "glacier" or "the glacier", when the context is clear, so the use of the full name will not make the text much longer. "FG" is used 12 times in the manuscript, sometimes up to three times in the same paragraph. The acronym "MGIC" for "mountain glaciers and ice caps" is also awkward and used much too often. The paragraph in lines 56 to 60 would, for example, be much easier to read without this uncommon acronym. Try to use as few acronyms as possible. In many cases, a minor rewording will eliminate the acronym and make the text flow better.

Thank you for that comment! We changed FG to Freya Glacier or simply to the glacier and we changed MGIC to (Greenland's) peripheral glaciers.

The use of hyphens ("-"), en-dashes ("–") and minus signs ("–") in composite words, negative numbers, number ranges and date ranges is inconsistent in many places. Use an en-dash or a proper minus sign for all negative numbers, also in superscripts such as "a^{-1}", and for all number and date ranges. Since you write "high-resolution DEM", you should probably also write "sea-level rise", and similarly for other compound adjectives (very many instances). The unit "meters water equivalent per year" should be written "m w.e. a^{-1}", not "m a^{-1} w.e. "

Thank you for that remark, we revised that accordingly.

**Description of Changes to Comments by Reviewer 2**

Bernhard Hynek, Daniel Binder, Michele Citterio, Signe Hillerup Larsen, Jakob Abermann, Geert Verhoeven, Elke Ludewig, Wolfgang Schöner. June 2024

RC: Reviewer Comment  AR: Author Response, description of changes in the revised version

In this paper, Hynek and colleagues present and analysis of very interesting data collected on Freya Glacier, one of Greenland's peripheral glaciers. They report annual glacier-wide mass balance observations from the glaciological method (2007 to 2022), as well as a geodetic survey between summer 2013 and summer 2021. They find close to equilibrium mass balance conditions, with a geodetic mass balance of 0.25 +/- 0.21 m w.e. over the eight years of survey. They link the observed pattern of elevation changes with the imprints of large avalanches that affected Freya Glacier in winter 2017/18, and that were investigated with an extensive ground penetrating radar survey.

The study is very interesting, and the data collected are of remarkable quality. This contribution is a long awaited one, as the topic of avalanche contribution to glacier mass balance remains poorly explored. Still I recommend major revisions, because there are two main points that would require some attention.

We appreciate the reviewer's thorough and insightful review of our work! In the following, we describe how we adressed them in a revised version of the manuscript.

**Major comments:**

1- Quantification of the avalanche contribution in the geodetic mass balance

Here I am sorry to be direct, and I might be wrong, but I am not sure that the method presented by the authors to separate the geodetic mass balance into areas that are affected by avalanches and areas not affected by avalanches is actually valid (L161-164). I do not understand why the mean elevation change of an areas that were not mapped as avalanche deposits in winter 2017/18 should not be affected by avalanches as well. If we write the kinematic relation for surface elevation changes, we get:

$$\partial h \partial t = b S \rho + w S - u S \partial h \partial x - v S \partial h \partial y$$

With h being the glacier surface, $bS\rho$ the surface mass balance normalized by density, $wS$ the vertical velocity, $uS$ and $vS$ the components of the horizontal velocity and $\partial h \partial x$ and $\partial h \partial x$ the components of the local slope. $wS - uS\partial h \partial x - vS\partial h \partial y$ is named the emergence velocity and $-uS\partial h \partial x - vS\partial h \partial y$ is named the advection of topography. This equation tells that the elevation change is the sum of surface mass balance and emergence velocity (or divergence of ice flux). The integral of the elevation change is equal to the integral of the mass balance only if done on a closed surface, which is not the case here, as there is a spatial continuity between the areas affected by avalanches and the areas not affected. As a simple example, one can imagine the deposit from an avalanche that would be advected by the flow and could change location within an elevation band, or change elevation band. Avalanches have also likely a non-local influence on the emergence velocity, simply because they lead to larger mass inputs, and thus larger ice flow. One solution to circumvent this difficulty is to calculate the distributed surface mass balance of the glacier from the elevation change map (e.g., Van Tricht et al., 2021; Vincent et al., 2021), but this required accurate knowledge of the glacier surface velocity, thickness, and to a lesser extent thermal regime.

I might also be wrong in my reasoning, and I think that the authors are absolutely right in their interpretation of the large impact of the winter 2017/18 avalanches, I would just be more careful on the quantitative side. Qualitative arguments are already quite strong regarding the persistence of snow three years after the event, and the good match between positive elevation changes and location of the deposits.

Thank you for this very relevant comment on the influence of glacier dynamics, namely the emergence velocity, which has the potential to alter the results of our quantitative assessment. We missed to tackle this question in

our manuscript. We appreciate also your suggestions how to solve this problem based on the recent work on that topic. The proposed methods to circumvent this problem need more input data, that might not be available in the needed quality, so we decided, not to use the elevation changes for a quantitative analysis of the mass contribution of avalanches. We base our quantification of the contribution of avalanches solely on the winter balance of 2018, where we have reliable data and we did not use the observed elevation change as a basis for that.

2- How frequent are the avalanches/how exceptional is winter 2017/18?

While the authors demonstrate clearly that the winter 2017/18 corresponds to a mass balance that is two sigma above the average and report that they are not aware of other large avalanches that affected Freya Glacier, I am not convinced that the glacier is not avalanche prone on "normal" years for some of its areas. In the hillshade from August 2013, there are signs of avalanche deposits or cones on the glacier surface, especially at the foot of the north east face, but also on the topographic right, around 600 m a.s.l. The authors could discuss whether the winter 2017/18 was exceptional compared with "normal" winters. One option would be to show other snow height maps to highlight the abnormal avalanche deposits. You could also investigate the climate records/reanalysis to assess the causes of this exceptional avalanche activity.

Thank you for this comment! We checked the available remote sensing data and found signs of avalanches also between 2012 and 2016, years with rather average winter accumulation. So, yes the glacier is prone to avalanches also in normal years, however to a far lesser extent. The avalanches visible on the glacier surface in 2014, 2015 and 2016 happened during the geodetic period 2013 - 2021, so they surely have contributed to the observed positive elevation changes in the respective regions. In the years after 2018 we could not find signs of new avalanches in the imagery, however they would have been harder to detect on top of the older avalanche deposits of 2018. The new figure 10 contains orthofotos, that show the extent of the 2012 and 2016 avalanches and the new figure 11 shows the end of summer snow cover in the mass balance period 2013-21.

To demonstrate the abnormal snow height distribution in 2018, we plotted distributed snow maps of 2008 and 2017, which were the only other years with a deailed GPR survey (new figure 9). We also plotted histograms of the individual GPR measurement points, sampled to 10 m resolution (see supplementary material) To quantify the impact of the extrapolation the unmeasured areas and the spline interpolation method we compared the arithmetic mean of all GPR data points to the area average of the snow height grid. While the difference in 2008 and 2017 is ~0.1 m, in 2018 the difference is 0.4 m. The larger difference in 2018 is due to the fact, that we estimated (introduced) snow depth values in unmeasured areas based on presumed avalanche flow pathes. This step is subjective, but might deliver the best estimate.

To demonstrate the exceptional meteorological situation in 2018 compared to other years, we added relevant climate data. We add continuous snow height data of AWS Freya Glacier, a temperature record from Zackenberg climate station and winter precipitation sums from the ERA5 reanalysis. We did not use snow height data and precipitation data from the climate station Zackenberg, as there are large data gaps in the relevant period in February 2018.

**Specific comments:**

L30-31: this sentence is not really clear to me. Do you suspect a bias in the data? Or do you observe a shift in the mass balance?

We changed the sentence to "Due to a gap in valid point observations caused by high accumulation rates and the COVID-19 pandemic the recently reported glacier-wide annual mass balance are rather crude estimates and show a negative bias in respect to the geodetic mass balance, which demands a thorough reanalysis of the glaciological time series."

There are limited links between the different paragraphs of the introduction. I think it should be possible to improve it a bit.

Thanks for this comment. We tried to make the transition smoother.

L72-73: the reference is an abstract from EGU. Consider removing it?

We have removed the reference.

There are many acronyms in the text. Consider spelling out Freya Glacier instead of its acronym. Same for the MGIC.

Thank you for this comment, this was also mentioned by reviewer 1. We changed this accordingly. Instead of MGIC we will use the term peripheral glaciers (of Greenland).

Supplement: I found the supplementary material by accident because it is not referred to in the text. I think it is important material, that demonstrates the very high quality of the two photogrammetric surveys, and it should be better emphasized (in L148 for example).

We added more figures to the supplementary material to describe technical details and we refer to the supplementary material in several places in the manuscript.

L112: I enjoyed very much looking at the automatic camera photographs! Thanks!

We are pleased to learn that you enjoyed it!

L134, 151-152 and 210-213: the correction applied to the geodetic survey is confusing because it is mentioned are three distinct locations, and inconsistent in some places (typo on the units on L212). I suggest to write from the beginning state that you apply a -0.60 m w.e. correction to the glacier wide mass balance, and potentially introduce the notations you use later on.

We changed that accordingly.

L143-144: how are the two DEMs/orthos merged? Consider providing more details about the elevation different on areas that are covered by both surveys.

Yes, there is an area in the middle of the glacier, which is covered by both surveys in 2013. Due to the poor coverage of that area (no near photo points and snow cover in diffuse illumination), surface reconstruction uncertainty is higher than in other parts. We added a figure in the supplementary material about the elevation differences in the overlapping area. Based on these elevation differences, the smoothness of the surface and an we drew a DEM border line. To get a smooth transition between the two DEMs we used weighted average values of both DEMs in a transition zone of + 100 m below and above the border line.

L154: "If feasible" suggests that you collected other GPR surveys of the snow thickness. It might be interesting to show some results from these surveys to highlight how winter 2017/18 is different from "average" winters.

We performed GPR surveys with a similar point observation density in spring of 2008 and 2017, in other years we have fewer snow depth point observations (see point observation number in new figure 11 (top panel). We show the interpolated snow maps of 2008, 2017 and 2018 in new figure 9.

L156-158: more details are needed about the avalanche deposit delineation. Which criteria do you use?

The delineation of avalanche deposits is a crucial step in the quantification of the mass contribution by avalanches. The delineation of the avalanche affected areas did not follow a strict, objective criteria, as this seemed to us hardly feasible and not very beneficial based on the available information. Along the GPR tracks we used a strong increase in GPR snow height for delineation. To complete the delineation in areas without GPR tracks we used a best estimate based on fotos of avalanche cracks, remnants of avalanches in the orthofoto of 2021 and above average local elevation changes together with likely avalanche flow pathes based on topography.

L158 [IMPORTANT]: what is the impact of this spline interpolation on the average snow thickness? On figure 8, it seems that the maximum snow thickness is not directly observed but extrapolated from the spline function. The pattern looks reasonable to me, as we expect maximum snow thickness close to the edges, but I think some lines about the uncertainty of this interpolation are needed.

We added more information on that, especially figure 9 and figure S5 (supplementary material). See also answer to your major comment 2.

L161: what is the value of the "bulk snow density"? Do you have multiple snow density estimates? Do you have density estimates of the avalanche deposits?

In spring 2018 we carried out one snow density measurement within a snow pit next to the AWS (see line 223). We removed the word bulk, as it is misleading. Snow density at the snow pit next to the AWS at stake 6 was 385 kgm$^{-3}$. There are no other snow density measurements in 2018. Particularly there are no observations of the snow density in the avalanche deposits. As the density of snow plays an important role in the calculation of the mass balance and the quantification of the contribution by avalanches, we added the data of the snow pit to the supplementary material and used the assumption of +5% and + 10% increase in mean snow density within the avalanche deposits.

L174-179: much more details are needed. First of all, it is not that usual to do fieldwork in spring to calculate annual mass balance. I imagine that there are some logistical constraints that explain this. You need to better explain how you find the ice surface and/or the horizon of the previous year. You also need to provide more details about the calculation of glacier-wide mass balance when only one or two stakes are found. The "statistical relationship" needs to be described, as well as the associate uncertainties.

In the revised version, we describe the mass balance evaluation and the associated uncertainties in more detail. We used a linear relationship between the mass balance at an "index stake" (at the AWS, stake 6, that is continuously measuring) and the glacierwide mass balance, derived in years with a lot of point observations. See also supplement for details.

L230: the current units for the stake measurements are m. This is a bit confusing and it would be better to use m w.e., as we are talking about surface mass balance here. The period is needed as well. On figure 5, the same comment applies: at stake location, the numbers correspond to elevation change (as suggested by the legend), or do they correspond to surface mass balance (as suggested by the text)? You could consider comparing the surface mass balance and elevation change at the stake location, this would give and idea of the impact of the dynamic.

We changed the units to m w.e. everywhere, except in figure 5, because here we show elevation changes in m and we think it is better comparable here. But we added a line in the figure caption to make clear, that the red numbers at the point locations refer to the height change measured at the stake. In addition we added a table to the supplement, where we compare ablation and elevation change at the stakes to give more information on the dynamic component.

L234: see my major comment 1, I doubt that the method can "predict" the glacier-wide mass balance without avalanches

Please see our answer to your major comment 1.

L243-244: repetition of L230

Thank you. We deleted the repetition.

L245: I find the unit m w.e. a-1 clearer that the unit ma-1 w.e. that is used here. Consider changing.
Discussion: the transition from the result section to the discussion is rather abrupt. Consider adding a few sentences to make a more seamless transition.

We changed that accordingly.

L249-262: this discussion is very interesting, but it could be expanded a bit by testing the impact of the different choices of density on the results?

Based on the suggestions of Reviewer 1, we used a mean density of 850 + 60 kgm-3 Huss (2013) here.

L257: issues with the citation formatting   Revised.

In general, the discussion could be sharpened and expended a bit. One aspect could be the climate context of Freya Glacier. I assume that there are very few climate record in the area, but it would be interesting to see whether the winter 2017/18 stands out in the climate record as particularly wet, and then cold or warm.

We added information on the climate context of Freya Glacier and the year of 2018.

L297-299: I agree with this statement, but it is never mentioned in the text before so it is a bit surprising to find it in the conclusion.

Thank you for that comment. We refer to this question earlier in the manuscript now.

The data availability statement could be more precise. The mass balance data are available through WGMS I assume? The DEMs or dh maps and snow depth maps could potentially be deposited on a repository.

Mass balance data are available through the WGMS and data until 2016 are also on pangaea.de (Hynek et al., 2014). We submitted the DEMs, Orthofotos and glacier outlines of 2013 and 2021 to pangaea.de. Until the data are available on pangaea.de we make them available under the following link: https://drive.google.com/drive/folders/1h_3TpU2Id12o1JUbUkMQ3hXRIxHDsjio?usp=sharing

References

Huss, M.: Density assumptions for converting geodetic glacier volume change to mass change, The Cryosphere, 7, 877–887, https://doi.org/10.5194/tc-7-877-2013, 2013.

Van Tricht, L., Huybrechts, P., Van Breedam, J., Vanhulle, A., Van Oost, K., and Zekollari, H.: Estimating surface mass balance patterns from unoccupied aerial vehicle measurements in the ablation area of the Morteratsch–Pers glacier complex (Switzerland), The Cryosphere, 15, 4445–4464, https://doi.org/10.5194/tc-15-4445-2021, 2021.

Vincent, C., Cusicanqui, D., Jourdain, B., Laarman, O., Six, D., Gilbert, A., Walpersdorf, A., Rabatel, A., Piard, L., Gimbert, F., Gagliardini, O., Peyaud, V., Arnaud, L., Thibert, E., Brun, F., and Nanni, U.: Geodetic point surface mass balances: a new approach to determine point surface mass balances on
glaciers from remote sensing measurements, The Cryosphere, 15, 1259–1276,
https://doi.org/10.5194/tc-15-1259-2021, 2021.

---

## Author Response (AR2)

**Answers to Review Comments #1**  (RC… Reviewer comment, AC… Author comment)

The revised manuscript is much easier to read than the original manuscript and largely without methodological flaws that need to be corrected. I have only a couple of comments and several editorial suggestions.

**Comments:** On page 11 (l. 417-319), it is stated: "The reason for the large bias of the glaciological record of -0.22 m w.e.a-1 in regard to the geodetic record needs further investigation using a distributed mass balance model."This comment does not properly reflect the typical magnitude of the bias of conventional mass-balance measurements. Andreassen et al. (2016) reported bias in conventional mass-balance records from 1ö Norwegian glaciers and found bias in the range -0.58 to +0.52 with respect to geodetic mass-balance estimates over several-year-long periods. Bias values exceeding 0.2 m w.e.a-1 in magnitude are in fact quite common judging from Andreassen's analysis. Jóhannesson et al. (2013) report bias of similar magnitude for glaciological mass-balance measurements of the Hofsjökull ice cap in Iceland.The sentence on page 11 should perhaps be replace by something like: "The magnitude of the bias is similar to bias estimates reported by Andreassen et al. (2016) for ten glaciers in Norway and therefore as such not unexpected (see also Zemp et al., 2013)."

Thank you for that comment, that puts our findings better in the context of the relevant literature: We changed that phrase according to your suggestion.

l. 175: "and ablation and accumulation was measured at several points". you must mean "summer and winter mass balance" or "seasonal mass balance" because ablation and accumulation are typically not directly measured in glaciological mass-balance measurements.

Thank you for that comment. Yes you are wright, we changed the wording accordingly.

**Minor and editorial comments:**
The English language and the general writing could be improved in many places.

We worked on improving the language and hope, that it has improved now. We considered all the minor and editorial comments below.

Examples:
l. 22: perhaps replace "... repeated photogrammetric surveys on 11 th - 18th August 2013 and on 28th - 31st July 2021 range from ..." with "... repeated photogrammetric surveys in August 2013 and July 2021 range from ..." (too much detail for an abstract?)
l. 24: rephrase: "A main imprint ..."
l. 26: rephrase: "when in addition to above average precipitation ..."
l. 29: rephrase: "Due to a gap in valid point observations ..." (was there no gap in the invalid observations?)
l. 31: rephrase: "which demands a thorough reanalysis ..."
l. 32; rephrase: "increases the likelihood of extreme snowfall events for individual years" perhaps simply say "increases the likelihood of extreme snowfall" ("events" adds no meaning, "extreme snowfall" tends to occur in specific years so "for individual years" also adds no meaning to this sentence)
l. 83: similarly "heavier single precipitation events" could perhaps also be replaced by "heavier precipitation"
l. 112: perhaps replace "(https://www.foto-webcam.eu/webcam/freya1/ and https://www.foto-webcam.eu/webcam/freya2/) (Freya Glacier Webcam 1, 2023; Freya Glacier Webcam 2, 2023)." by "Freya Glacier Webcam 1: https://www.foto-webcam.eu/webcam/freya1 and Freya Glacier Webcam 2: https://www.foto-webcam.eu/webcam/freya2" (and eliminate the corresponding references from the reference list; these are not proper references)

l. 146-149: perhaps replace "Georeferencing of the two final DEMs is based on all respective GCPs, a co-registration of the DEMs (Nuth and Kääb, 2011) was not carried out, as the overlapping area on stable terrain outside the glacier is too small. However the small overlapping and supposed stable area was used to calculate error statistics of the two DEMs (see supplement)." by "As the georeferencing of the two DEMs is based on a large number of GCPs, co-registration of the DEMs (Nuth and Kääb, 2011) was not needed. Elevation differences in small, overlapping stable, ice-free terrain had mean bias of ? m and RMS of ? m (see supplement)."

This list is only a few examples from a much larger set of sentences that might be improved.

References:
Jóhannesson, T., Björnsson, H., Magnússon, E., Guðmundsson, S., Pálsson, F., Sigurðsson, O., Thorsteinsson, Th., Berthier, E. (2013). Ice-volume changes, bias estimation of mass-balance measurements and changes in subglacial lakes derived by lidar mapping of the surface of Icelandic glaciers. Annals of Glaciology, 54(63), 63-74. doi: 10.3189/2013AoG63A422
Andreassen, L. M., Elvehøy, H., Kjøllmoen, B., and Engeset, R. V. (2016). Reanalysis of long-term series of glaciological and geodetic mass balance for 10 Norwegian glaciers. The Cryosphere, 10, 535–552, doi: 10.5194/tc-10-535-2016

**Answers to Review Comments #2**  (RC… Reviewer comment, AC… Author comment)

This review considers the revised manuscript from Bernhard Hynek et al., entitled "Accumulation by avalanches as significant contributor to the mass balance of a High Arctic mountain glacier". The authors addressed the comments from my previous review in a satisfactory way, and I thank them for the time and efforts dedicated to implementing these changes. I have only one general comment that is more a thought on this topic and can easily be addressed. All the other ones are technical.

We again want to thank the reviewer for his constructive contribution and his time put into our manuscript!

**General:** on the orthoimage of 2021, all the patches of snow are considered as the remnants of the 2018 avalanches. Looking at figures 10 and 11, it seems that some of these areas are affected by avalanches more of less every year. Even though it is certain that these areas were affected by the 2018 avalanches, they might have also been affected afterwards, and hence the snow is not necessarily a remnant of 2018. This reasoning can also apply to the elevation change map: if these areas are regularly affected by avalanches, they act as localized accumulation areas with a positive surface mass balance (as visible on Fig. 10 and 11 with snow remaining at the end of the ablation season). We thus expect these areas to adjust in a dynamic way, with increased surface slope that leads to locally larger mass fluxes, as shown by the surface protruding at the foot of the large couloir on the right side of the glacier tongue (already visible on the 2013 hillshade, fig. 3). I just suggest to remove the references to the remnants of the 2018 avalanches, a call them as "avalanche affected areas".

Thank you for that comment! We fully agree and changed the wording as suggested in L 224 and in the caption of Fig. 11.

We adressed all the technical corretions and suggestions below.

**Technical Comments:**
L23: « Somewhat surprisingly » is weird -> the average elevation change is positive and hence the MB
Citations au milieu de phrases (L65, 68)
L67: typo
L75: no need of the acronym
L94: typo -> snow depth
L104: explain the origins of the different spellings
We added the following footnote: [1] According to the Language Secretariat of Greenland (Oqaasileriffik.gl) the official name is spelled as Frejagletsjer (formerly Frejagletcher). While (Ahlmann, 1946) used Fröya Glacier, in (Higgins, 2010) the glacier was also spelled as Fröjabreen, Frøya Glacier and Fröya Glacier. In recent scientific literature (e.g. Schöner et al., 2009) the spelling Freya Glacier has been used.
L109: update the reference to WGMS
L144: "afterwards" typo
L159: Huss 2013 (not "et al.")
L168: rather informal "to get a good picture of"
L169: impact of the spline function?
L170: add ref to the supplement
L171: a spatial average of what?
L175: are -> were
L192 and elsewhere: "foto" -> picture, image
L197: units
L204 "snowed in" -> familiar; "buried in snow"
L242: section title is strange. Maybe change for "Winter 2018 and avalanches"
L245-246: unclear sentence with a problem of date formatting
L258: "a"->"the"

L260: are you discussing surface mass balance, as written in the text, or elevation changes, as shown on the figure 6b
We are discussing surface mass balance at the stakes. We clarified this by the reference "(see stake readings in Fig 6b and Table S1)."

L312: remind the contribution of avalanches to the winter mass balance in percentage
L293-295: this is a very interesting point. I was just wondering how do you estimate the surface mass balance in the accumulation area from the spring measurements only? Do you manage to observe the previous year horizon to assess how much mass was lost during summer?
In some years, especially when winter snow depth was low, this was done by digging down to the previous years' horizon. In some years this was not feasible due to limited time on the glacier. We recognize, that this is a limitation in the glaciological mass balance observation at Freya Glacier and will focus more on that in the future.

Fig. S1: consider using another colorscale because the rainbow color scale is not appropriate for colorblind people and induces misperception of the spatial patterns

---

## Author Response (AR3)

Dear Editor,

Please find our point-by-point response letter to your comments below.

Kind regards, Bernhard Hynek
* * *
Editor Comment – Author Comment
* * *
Before a final decision can be made, some corrections are still needed. In particular, one of the comments of reviewer#2 was not fully addressed, regarding the presence of the 2018 avalanche snow at the surface in 2021. This could also originate from avalanches in winter 2019, 2020 and/or 2021. This may lead to modifications at several instances in the text.

We changed the wording *remnants of avalanches 2018* to *avalanche affected areas* in several instances in the text to make it less specific and added some lines in the discussion section, where we argue, that some of these areas are most likely remnants of 2018.

L2: "High arctic" make the study generic but I feel it would be good to be more precise about the location. An option would be "...mass balance of a peripheral glacier of Greenland" or " a mountain glacier at the periphery of Greenland"

We agree. We suggest to change the title to: *Accumulation by avalanches as significant contributor to the mass balance of a peripheral glacier of Greenland*

L24: This number and its difference to 1.33 m w.e. just above is enigmatic to the reader until he understands, line 160, that a seasonal offset to the end of the melt season is involved and corrected. I would include in parenthesis a very short statement about this seasonal correction.

Thanks for that comment. We changed the phrase to: *After applying a seasonal correction of -0.6 + 0.05 m w.e. the geodetic mass balance over the entire eight-years period (2013/14 - 2020/21) is found to be 0.73 + 0.22 m w.e.*

L24: This point has been raised by reviewer #2 but not really addressed I think. How can the authors be certain that the snow of the 2018 avalanches is visible at the surface? This could also come from avalanches in winter 2019, 2020 and/or 2021.

Thanks for that comment. Off course we cannot be certain about that, and when we made that statement, we were actually referring only to the two big avalanches in the middle of the glacier (green circle in Fig. 11), which originated from opposite sides and travelled almost all the way through the other side of the glacier.

In the case of these two avalanches, we have good reasons to believe, that these are the remnants of 2018 because
- the extent of the snow cover 2021 matches very good the avalanche areas of 2018 based on the GPR survey (we marked that in Fig.10)
- winter accumulation in all 3 years after 2018 was far below average and was therefore very unlikely to produce avalanches of that size
- we have no signs of big avalanches in 2019, 2020 and 2021 there in imagery of the automatic cameras
- we can see signs of small avalanches on top of the big ones (of 2018)

Here, in the abstract, we changed the statement about the surface in 2021 to the following sentence: *Remote sensing data show, that Freya Glacier is prone to avalanches also in other years, but to a lesser spatial extent.*

At several instances in the text, we changed *remnants for the 2018 avalanches* to *avalanche affected areas.*

In the discussion, we added the following text:

*The avalanche cycle of 2018 was outstanding in regard to the mass input and the glacier area affected. However, avalanches seem to be a persistent feature on Freya Glacier, as their deposits are visible almost every year.  It is difficult to date these avalanches and estimate their frequency, as older avalanche deposits might get covered by new ones. In case of the two big avalanches in the middle of the glacier which originated from opposite sides and travelled almost all the way through the other side of the glacier in 2018 (green circle in Fig. 11), we have strong evidence, that their remnants are still visible in the orthophoto of 2021, more than three years after the incident. On the one hand it takes a few ablation seasons to melt avalanche snow up to 8 meters thick, particularly, if that snowpack is located in a rather flat area on the glacier, where it is more likely to get densified by retention of meltwater. On the other hand, winter mass balances in the following years were below average (Fig. 12) and therefore unlikely to produce avalanches of this size. While avalanche deposits are easy to identify in the ablation area or in rather negative mass balance years, their presence and extent remain equivocal in the upper firn area or less negative years.*

L29: add "(burried stakes)" to suggest why high accumulation rates leads to missing years/Gaps.

Thanks for that comment. We added *(buried stakes).*

L44: 11% is not their sea level fraction but contribution to the overall loss of Greenland. So, move this number after "Greenland", otherwise this is ambiguous.

We added that accordingly.

L112: I checked the website 6/9/24 and it seems the webcams were offline since 28 April 2024... Delete the link or provide an explanation.

Yes, the webcams are offline due to a technical problem, but most likely will be online again after the next fieldwork in April 2025. Via the link readers can also use the calendar function to look at images in previous years. So, we would like to leave the link here, but we added the following footnote: *Due to technical problems the webcams are offline since April 2024, but older camera images can still be found here.*

L172: Unclear. Do you address "sampling" here or "material density"? Clarify.

We wanted to say that the spatial coverage of the glacier surface with GPR tracks was comparable in those years. (While in other years we did only snow depth probing, with a far lesser number of observations and a reduced spatial coverage.) We changed the wording to: *Similar GPR snow surveys with comparable spatial coverages have been carried out in spring 2008 and 2017, while in other years a reduced sample network was used.*

L176: Typo glacierwide: here and elsewhere  Thanks, we corrected it everywhere in the manuscript!

L177: a reference for this error.

We reference (Pulwicki et al., 2018) here, they found typical winter balance uncertainties ranging from 0.03 to 0.15 m w.e. As our point observation density is very high, we estimated an uncertainty of 0.05 m w.e., which is in the lower range of their estimate.

L198: I guess these values are difficult to back up. But which value did you exactly used? 5 or 10%? Right now, it is not clear enough for the reader why two values are listed. Later in the article I see that this is values for sensitivity tests. Present them this way here also.

Thank you for that comment, that was ill-conceived and inconsistent. We changed the reasoning throughout the paper to the following:  We use a 5% increase of snow density within the avalanche deposits (in regard to the undisturbed snow pack) as our best guess to account for compaction due to overburden pressure and avalanche deposition. And we use a 10% increase as an upper boundary, which only appears in the error of the mass balance contribution of 0.35 + 0.04 m w.e. in line 262.

The phrase reads now:

*To convert snow heights into snow water equivalent, we used the mean snow density of 385 kg m$^{-3}$ (measured in the snow pit next to the automatic weather station) for areas that are not influenced by avalanches. As snow density typically increases with snow depth and avalanche deposits have higher snow densities than the undisturbed snow pack we used higher snow densities for the avalanche deposits: a 5% increased snow density (404.25 kg m$^{-3}$) as a best guess and 10% increase (423.5 kg m$^{-3}$) which we interpret as an upper boundary.*

L206: Using the exact same snow height values or using a correction factor to account for the different setting? I can imagine that the snow heights are not exactly the same on these two glaciers. I see this is detail in the supplement. Add a sentence indicating that the AP Olsen ice cap are corrected and refer to the supplement.

We added: *The data gap of 2.5 months was reconstructed using snow height data from the main weather station at A. P. Olsen Ice Cap (Larsen et al., 2023; Greenland Ecosystem Monitoring, 2020a), which has a continuous record in 2018 (see supplement, Fig. S6).*

L225: maybe "sampling" ?  We changed that accordingly.

L227 "still" not really needed here (as the 2018 avalanches have not been described yet)           Deleted.

L231 This is shown in Figure 6b. Authors should check carefully the figure numbers and make sure they are called sequentially in the text. Make sure all figures are referred to in the text.

Thank you. We corrected that and checked all Figure and Table references to be in the right order.

L231 It seems that a small part close to the glacier front is not covered also.

Yes, we forgot to mention this part. We changed the wording to ... *missing only some parts in the upper accumulation zones and a small debris covered area next to the glacier snout.*

L243 You could provide the equivalent annual rate in parenthesis. We added *(0.09 m w.e. a⁻¹).*

L261 See my main comments above (echoing Rev#2 comment). Could be the trace of avalanches in 2019, 2020 or 2021 also.

We deleted the text here and added some lines in the discussion section instead, as stated above.

L262: See my above comment that figures should be numbered sequentially in their order of referencing in the text. Corrected.

L265 The title of the section does not match with the content of the paragraph that discuss the visibility of the avalanche on the glacier surface...

We changed the section title to: *Visibility of avalanches on the glacier surface.*

L267 Authors would need to annotate (circles?) these photos (Fig. 10 and 11) to make it clear to the readers where the deposits are visible. We outlined all visible avalanches in Fig. 10 and added circles to the main avalanche zones within the ablation area in Fig. 11.

L271 No section 4.6 Corrected

L273 Fig 12... Corrected. This reads as if the availability of a DEM would impact the mass balance... To be rewritten.

We changed the sentence to:

*Prior to 2013, all annual mass balances were negative, with 2013 having the most negative mass balance on record so far. Higher winter mass balances between 2014 and 2018 can be associated with some positive annual mass balances in that period, while after 2019 drier winters facilitated again negative annual mass balances.*

282 How does this sentence relate to the discussion? Clarify the not so obvious link. In fact I had a hard time following the logic of the whole paragraph. Can you try to make this logic more obvious to the reader? Do you want to justify the Huss density assumption? Justify the 5 to 10% higher density?

Thank you for the comment, that this is hard to understand. Yes, we wrote this to justify both the 5 to 10% higher density assumption of the avalanche snow and the Huss density assumption. We changed the wording of the paragraph and hope, that it has become clearer now.

L 297: It is "internal accumulation" if these deep layers are below the last summer horizon. It is what you mean? Clarify

Yes, that is what we meant. We changed the phrase to: *Another likely reason for the bias between the glaciological and geodetic mass balance is the already mentioned unknown magnitude of meltwater retention by refreezing within deeper firn layers.*

L299 I think "necessary" is best here. Thank you. We changed that accordingly.

L316 I miss a clear demonstration that the deposits visible in 2021 are really from 2018 (as said above and noted by rev#2). This point needs to be addressed. See our statement above.

L321 exponent Corrected

L324 I suggest to add: "leading to internal accumulation" Thank you. We changed that accordingly.

505 Typos and grammar Corrected

508 no cap letters Corrected

520 Fig 8c is almost not discussed in the text. I think it is important to back up the statement of a regional phenomenon.

We changed the reference to Fig. 8c in the text to: (L248)

*In the winter of 2017/2018, a series of low-pressure systems between the southern tip of Greenland and Iceland transported warm and moist air masses to the East Coast of Greenland with frequent snowfall leading to above average winter precipitation sums along large parts of the East Coast (Fig. 8).*

522 Fig 9 caption typo bolt changed to bold

525 Fig 10 See comment in the main text to circled the visible avalanche deposits. We added outlines of visible avalanche

deposits to the figure

L530 Fig 11 See comment in Fig. 10  We added circles to the figure.

L537 maybe clarify that this results from extensive GPR survey? Reader can be surprised by this number.

To clarify this, we added: *The high number of point observations in winter 2017/18 corresponds to 10 m along track mean snow depth values of the extensive GPR survey.*